# Meta-learning Structure-Preserving Dynamics

**Cheng Jing** [1]  **Uvini Balasuriya Mudiyanselage** [1]  **Woojin Cho** [2]  **Minju Jo** [3]  **Anthony Gruber** [4]  **Kookjin Lee** [1]

## Abstract

Structure-preserving approaches to dynamics discovery have demonstrated great potential for modeling physical systems due to their use of strong inductive biases, which enforce key features such as conservation laws and dissipative behavior. However, these models are typically trained on a per-configuration basis, requiring explicit knowledge of system parameters and costly retraining when these parameters vary. While meta-learning provides a potential remedy, optimization-based approaches can suffer from limited generalizability. Motivated by recent advances in modulation-based learning aimed at mitigating these drawbacks, we systematically investigate the use of modulation techniques in learning conservative dynamical systems. We study a range of existing modulation strategies alongside newly proposed variants, integrating them into a Hamiltonian learning framework without requiring an explicit system parameterization. Through extensive experiments on benchmark problems, we demonstrate that modulation-based meta-learning enables accurate few-shot adaptation, achieving robust generalization across parameter space without compromising the conservation of key invariants responsible for the dynamics.

## 1. Introduction

Scientific machine learning (SciML) has emerged as a powerful paradigm for modeling complex physical systems (Baker et al., 2019), enabling the discovery and prediction of dynamic behavior in fields ranging from fluid dynamics (Karniadakis et al., 2021; Brunton et al., 2020) to materials science (Butler et al., 2018). Learning dynamics

from data is essential when physical laws are unknown, partially known, or when direct simulation is computationally prohibitive. While physics-informed neural networks and related approaches have shown promise by incorporating physical biases into the learning process (Raissi et al., 2019), their use of soft constraints typically does not enforce the preservation of structural properties exactly, especially when the learned models are queried for dynamical prediction outside of their training distribution. This can lead to poor generalization performance and the substantial violation of fundamental physical principles encoded in mathematical conservation laws. As a result, there is growing interest in developing tailored machine learning models that faithfully preserve the mathematical structure of physical systems.

Recent work has introduced structure-preserving neural networks that embed physical inductive biases directly into their architectures. Hamiltonian neural networks (HNNs) (Greydanus et al., 2019; Toth et al., 2020) and Lagrangian neural networks (Lutter et al., 2018; Cranmer et al., 2020) conserve a notion of energy by modeling the system through its underlying symplectic or variational structure. Principled extensions of these models, such as port-Hamiltonian neural networks (Zhong et al., 2020; Desai et al., 2021), generalized Onsager principle-based networks (Yu et al., 2021) and metriplectic neural networks (Hernandez et al., 2021; Lee et al., 2021a; Gruber et al., 2023; 2025), have been proposed to handle dissipative systems, which include phenomena such as energy exchange and entropy production. While these structure-preserving approaches offer improved physical fidelity, they generally require complete knowledge of the system's parameters. Since parametric variation in the system (through, e.g., quantities like mass and damping strength) may not be fully known, these networks are trained in a parameter-specific manner, and a new model must be trained for each parametric configuration. This retraining process creates significant inefficiency in practice, sometimes leading to delayed adoption of structure-preserving methods in "many-query" applications.

Conversely, many real-world dynamical systems exhibit parametric variations that are predictable and intuitive. For example, the motion of a series of pendula may depend smoothly on pendulum masses and rod lengths. Some of the Hamiltonian learning literature has explored meta-learning

[1]School of Computing and Augmented Intelligence, Arizona State University, USA [2]TelePIX Co., Ltd., South Korea [3]Sorbonne Université, CNRS, ISIR, France [4]Center for Computing Research, Sandia National Laboratories, USA. Correspondence to: Kookjin Lee <kookjin.lee@asu.edu>.

*Proceedings of the 43rd International Conference on Machine Learning*, Seoul, South Korea. PMLR 306, 2026. Copyright 2026 by the author(s).

approaches that learn initializations of model weights based on system parameters (Lee et al., 2021b; Song & Jeong, 2024), enabling faster adaptation to new problem configurations. These methods typically build on established meta-optimization strategies like model-agnostic meta-learning (MAML) (Finn et al., 2017), which aim to produce models capable of generalizing across a family of tasks. However, they also require gradient-based updates to high-dimensional model parameters, which can be expensive and potentially unstable. Moreover, there has been no thorough comparison of these optimization-based methods against the more recent modulation-based meta-learning techniques such as (Dupont et al., 2022), which can alleviate these issues through targeted or low-rank parameter updates.

In view of this, we investigate a modulation-based meta-learning framework, sidestepping the issues with explicit optimization given new parameter configurations by directly conditioning the learnable dynamics model on low-dimensional latent variables. Inspired by recent advances in implicit neural representations (INRs) (Sitzmann et al., 2020; Tancik et al., 2021), where modulation is used to generalize across scenes or signals (Dupont et al., 2022), we adapt and extend these techniques to the setting of structure-preserving dynamics. We also propose a low-rank modulation technique suitable for Hamiltonian systems, and benchmark its performance against common optimization-based and modulation-based meta-learning strategies.

To summarize, our main technical contributions include:

- exploration of modulation-based techniques for the meta-learning of provably structure-preserving dynamics with unknown parametric dependence,
- development of two novel modulation techniques with improved expressiveness over existing approaches,
- extensive experimentation on benchmark problems consisting of Hamiltonian (energy-conserving) systems, with additional experimentation on metriplectic (dissipative) systems.

## 2. Technical Background

We begin by describing established parameterization techniques for preserving important structure governing energy conserving and dissipative systems.

### 2.1. Hamiltonian Mechanics and Hamiltonian Neural Networks

Hamiltonian systems are dynamical systems on phase space which arise from Hamilton's least-action principle. The state of these systems is canonically described by a set of generalized coordinates and their associated momenta, denoted as $\boldsymbol{q} = [q_1, ..., q_n]$ and $\boldsymbol{p} = [p_1, ..., p_n]$, respectively. The dynamics are then defined by the symplectic gradient of

the Hamiltonian function(al) $\mathcal{H}(\boldsymbol{q}, \boldsymbol{p})$ on phase space, which represents the total energy of the system, i.e., the sum of the kinetic and potential energies. Precisely, the (canonical) Hamiltonian dynamics are defined by

$$\frac{\mathrm{d}\boldsymbol{q}}{\mathrm{d}t} = \frac{\partial \mathcal{H}}{\partial \boldsymbol{p}}, \quad \frac{\mathrm{d}\boldsymbol{p}}{\mathrm{d}t} = -\frac{\partial \mathcal{H}}{\partial \boldsymbol{q}}. \tag{1}$$

A key property of the evolution defined by Eq. (1) is that an isolated Hamiltonian system is guaranteed to conserve the total energy along solution trajectories. This follows from the vanishing of the dissipation rate: $\frac{\mathrm{d}\mathcal{H}}{\mathrm{d}t} = \left(\frac{\partial \mathcal{H}}{\partial \boldsymbol{q}}\right)^{\mathsf{T}} \frac{\mathrm{d}\boldsymbol{q}}{\mathrm{d}t} + \left(\frac{\partial \mathcal{H}}{\partial \boldsymbol{p}}\right)^{\mathsf{T}} \frac{\mathrm{d}\boldsymbol{p}}{\mathrm{d}t} = \left(\frac{\partial \mathcal{H}}{\partial \boldsymbol{q}}\right)^{\mathsf{T}} \frac{\partial \mathcal{H}}{\partial \boldsymbol{p}} - \left(\frac{\partial \mathcal{H}}{\partial \boldsymbol{p}}\right)^{\mathsf{T}} \frac{\partial \mathcal{H}}{\partial \boldsymbol{q}} = 0.$

The definition of Hamiltonian systems in terms of a single scalar potential function $\mathcal{H}$ suggests a simple and natural parameterization for machine learning. HNNs (Greydanus et al., 2019) parameterize the Hamiltonian function with a neural network $\mathcal{H}_{\Theta}(\boldsymbol{q}, \boldsymbol{p})$ consisting of learnable model parameters $\Theta$. This choice allows HNNs to be trained by minimizing the loss

$$\mathcal{L}_{\text{symp}} = \left\| \frac{\mathrm{d}\boldsymbol{q}}{\mathrm{d}t} - \frac{\partial \mathcal{H}_{\Theta}}{\partial \boldsymbol{p}} \right\|_2^2 + \left\| \frac{\mathrm{d}\boldsymbol{p}}{\mathrm{d}t} + \frac{\partial \mathcal{H}_{\Theta}}{\partial \boldsymbol{q}} \right\|_p^2, \tag{2}$$

which we denote by the "symplecticity loss". A similar argument to the above guarantees that solution trajectories to Hamilton's equations will conserve the approximate total energy $\mathcal{H}_{\Theta}$.

### 2.2. Metriplectic Mechanics and GENERIC Neural Networks

Despite the tremendous utility of the Hamiltonian formalism in describing idealized limits of dissipative phenomena, e.g., the Euler equations as a zero-viscosity limit of Navier–Stokes, many systems have nontrivial dissipative effects. While there are several theoretical frameworks which lead to useful and general descriptions of dissipation, this work considers systems satisfying the General Equation for Non-Equilibrium Reversible-Irreversible Coupling (GENERIC) formalism. Also known as metriplectic systems (Morrison, 1986; 2009), GENERIC systems are complete thermodynamical models with simultaneous guarantees on the conservation of free energy and the generation of entropy.

Analogous to Hamiltonian systems and HNNs for energy-conserving dynamics, the GENERIC/metriplectic formalism provides a structured representation for systems with joint conservative and dissipative effects. This work leverages the GENERIC neural network (GNN) parameterization developed in (Lee et al., 2021a) to model dissipative dynamics; technical details are deferred to Appendices B–C.

**A system of pendula**

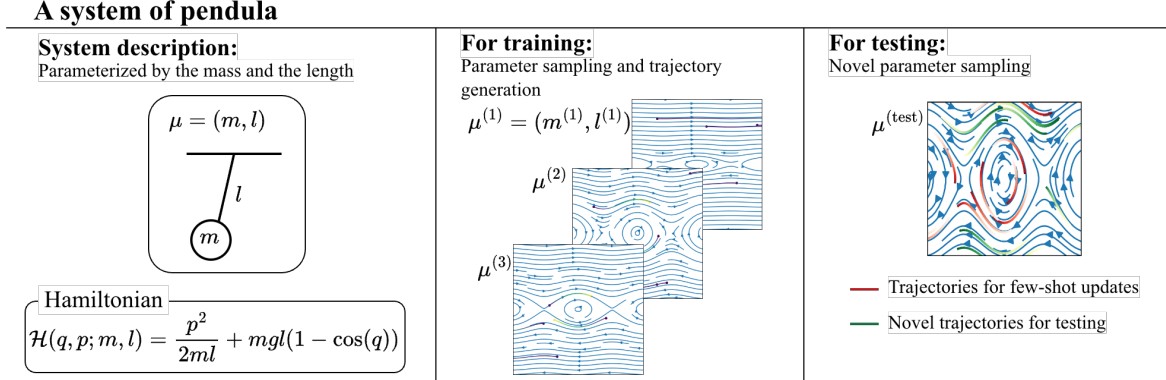

*Figure 1.* Overview of the problem setup for meta-learning structure-preserving dynamics.

## 2.3. Modulation for Neural Networks

Modulation refers to a class of techniques, widely explored in convolutional neural networks (Perez et al., 2018) and recurrent neural networks (Birnbaum et al., 2019), that condition on task- or signal-specific information. Also known as "feature-wise linear modulation (FiLM)", this idea has been used to great effect in implicit neural representations (INRs) (Sitzmann et al., 2020; Fathony et al., 2020; Dupont et al., 2022; Cho et al., 2023) and neural radiance fields (NeRFs) (Park et al., 2019). However, these approaches are not inherently generalizable or memory efficient, as they require learning and storing separate parameters for each signal, which becomes impractical for large or continuously varying datasets encountered in dynamics modeling.

Latent modulation addresses these limitations (Dupont et al., 2022) by first mapping each signal to a compact latent vector. Then, a learned function such as a hypernetwork, parameterized by a small multilayer perceptron (MLP), is used to generate layer-wise modulation parameters. For example, latent shift modulation modifies the MLP layer $h \mapsto \sigma(W^{(\ell)}h + b^{(\ell)})$ to $h \mapsto \sigma(W^{(\ell)}h + b^{(\ell)} + s^{(\ell)}(z))$, where $s^{(\ell)}(z) = f_{\text{hyper}}(z^{(k)}; \phi)$ is the output of a hypernetwork. This formulation enables adaptation across a wide range of signals using a shared embedding space while supporting interpolation, few-shot learning, and improved scalability (Dupont et al., 2022; De Luigi et al., 2023; Zhou et al., 2023).

## 3. Methods

We now detail the problem setting under consideration, including descriptions of the proposed techniques for meta-learning structure preserving dynamics and associated algorithms for training/testing the proposed models.

## 3.1. Problem Setup

Consider a parameterized dynamical system, $\frac{d\boldsymbol{x}}{dt} = \boldsymbol{f}(\boldsymbol{x}; \boldsymbol{\mu})$, where $\boldsymbol{x}$ denotes the state variables of the system and $\boldsymbol{\mu}$ denotes a set of potentially unknown parameters. The velocity vector field $\boldsymbol{f}$ could describe either conservative dynamics, such as Hamiltonian, or dissipative dynamics, such as GENERIC. The set of parameters $\boldsymbol{\mu}$ is variable across different systems; for example, a system of pendula can be characterized by variable masses and lengths $\boldsymbol{\mu} = (m, l)$, while a Duffing oscillator can be characterized by varying stiffness and nonlinearity $\boldsymbol{\mu} = (\alpha, \beta)$. We are interested in learning these parameterized dynamics with a neural network $\boldsymbol{f}_\Theta(\boldsymbol{x}, \boldsymbol{\mu}) \approx \boldsymbol{f}(\boldsymbol{x}, \boldsymbol{\mu})$ while preserving important structures in the learned model.

We consider one set of governing equations at a time, e.g., a pendulum characterized by the parameters $\boldsymbol{\mu} = (m, l)$ will be treated as different from a Kepler system with parameters $\boldsymbol{\mu} = (m_1, m_2)$. Given such a system and a range of input parameters, we assume that multiple trajectories are generated from variable initial conditions for each realization of the input parameters. For example, the data sampling procedure in the case of Hamiltonian systems is the following. First, a set of parameter instances $\{\boldsymbol{\mu}^{(k)}\}_{k=1}^{n_\mu}$ is randomly sampled within a certain range, yielding a corresponding set of system Hamiltonians $\mathcal{H}^{(k)}(\boldsymbol{q}, \boldsymbol{p}) = \mathcal{H}(\boldsymbol{q}, \boldsymbol{p}; \boldsymbol{\mu}^{(k)})$. We next collect $n_\mathcal{T}$ trajectories of length $n_{\text{seq}}$ by varying initial conditions randomly in a certain domain, denoting the $i$-th trajectory collected from the $k$-th Hamiltonian system by $\mathcal{T}^{(k,i)} = \left((\boldsymbol{q}_1^{(k,i)}, \boldsymbol{p}_1^{(k,i)}), \dots, (\boldsymbol{q}_{n_{\text{seq}}}^{(k,i)}, \boldsymbol{p}_{n_{\text{seq}}}^{(k,i)})\right)$. Once all trajectories have been generated by iterating through the $n_\mu$ systems, they are split into training $\mathcal{D}_{\text{train}}$ and testing $\mathcal{D}_{\text{test}}$ sets with cardinality $n_\mu^{\text{train}}$ and $n_\mu^{\text{test}}$ such that $n_\mu = n_\mu^{\text{train}} + n_\mu^{\text{test}}$. See Figure 1 for an illustration.

## 3.2. Modulation-based Parameterization

Given a Hamiltonian system (for example) and the data just described, we seek an effective representation $\tilde{\mathcal{H}}(\boldsymbol{q}, \boldsymbol{p}; \boldsymbol{\Theta}^{(k)}) \approx \mathcal{H}(\boldsymbol{q}, \boldsymbol{p}; \boldsymbol{\mu}^{(k)})$ for each system Hamiltonian, where $\tilde{\mathcal{H}}$ denotes a neural network and $\boldsymbol{\Theta}^{(k)}$ denotes its (potentially numerous) model parameters. To avoid the heavy retraining cost seen in standard Hamiltonian learning methods, these model parameters $\boldsymbol{\Theta}^{(k)} = \boldsymbol{\Theta}_{\text{base}} \cup \boldsymbol{\Theta}_{\text{indv}}^{(k)}$ are separated into two parts: a set of base parameters $\boldsymbol{\Theta}_{\text{base}}$ and a set of individual parameters $\boldsymbol{\Theta}_{\text{indv}}^{(k)}$. The base parameters are meta parameters shared across all system Hamiltonians $\{\mathcal{H}^{(k)}\}_{k=1}^{n_\mu}$, while the individual parameters are realization-specific to each Hamiltonian $\mathcal{H}^{(k)}$. This allows for effective "pre-training" of the common parameters $\boldsymbol{\Theta}_{\text{base}}$ used to capture bulk characteristics shared between all Hamiltonians, along with computationally inexpensive "fine-tuning" updates using $\boldsymbol{\Theta}_{\text{indv}}$ which are calibrated to finer features of each specific system.

**Network architecture** The basic architecture underlying the proposed modulation strategy consists of MLPs, whose operation in the $\ell$-th layer can be succinctly represented as $\boldsymbol{h} \mapsto \sigma(\boldsymbol{W}^{(\ell)}\boldsymbol{h} + \boldsymbol{b}^{(\ell)})$, where $\boldsymbol{h}$ denotes the hidden state vector, $\boldsymbol{W}^{(\ell)}$ and $\boldsymbol{b}^{(\ell)}$ denote the weights and biases at layer $\ell$, respectively, and $\sigma$ denotes a nonlinear activation function. In the case of latent modulation, the individual model parameters, $\boldsymbol{\Theta}_{\text{indv}}^{(k)}$, are inferred from a hyper-network $\boldsymbol{f}^{\text{hyper}}(\boldsymbol{z}^{(k)}; \boldsymbol{\phi})$, where $\boldsymbol{\phi}$ denotes the model parameters of $\boldsymbol{f}^{\text{hyper}}$ and $\boldsymbol{z}^{(k)}$ denotes a latent code. Each latent code is expected to contain salient features of the corresponding neural network Hamiltonian e.g., $\boldsymbol{z}^{(k)}$ for $\tilde{\mathcal{H}}^{(k)}$.

**Modulation techniques** For an MLP with weights and biases $\{\boldsymbol{W}^{(\ell)}, \boldsymbol{b}^{(\ell)}\}_{\ell=1}^{L}$, we propose two novel modulation techniques which apply layer-wise corrections to these parameters. The first approach is inspired by the low-rank adaptation (LoRA) methods (Hu et al., 2022), which have been used to great success in large language models. The second method is more parameter-efficient, resembling the singular value decomposition (SVD) of a matrix. Here and in the following, they are denoted by latent multi-rank (MR) and latent SVD-like modulation, respectively. At layer $\ell$, MR is defined as

$$\boldsymbol{h} \mapsto \sigma\left(\left(\boldsymbol{W}^{(\ell)} + \boldsymbol{U}^{(\ell,k)}\boldsymbol{V}^{(\ell,k)\top}\right)\boldsymbol{h} + \boldsymbol{b}^{(\ell)} + \boldsymbol{s}^{(\ell,k)}\right), \tag{3}$$

where $\boldsymbol{U}^{(\ell,k)}, \boldsymbol{V}^{(\ell,k)} \in \mathbb{R}^{w_\ell \times r}$ with the width $w_\ell$ and

$$\left(\cup_{\ell=1}^{L}\boldsymbol{U}^{(\ell,k)}, \cup_{\ell=1}^{L}\boldsymbol{s}^{(\ell,k)}, \cup_{\ell=1}^{L}\boldsymbol{V}^{(\ell,k)}\right) = \boldsymbol{f}^{\text{hyper}}(\boldsymbol{z}^{(k)}; \boldsymbol{\phi}). \tag{4}$$

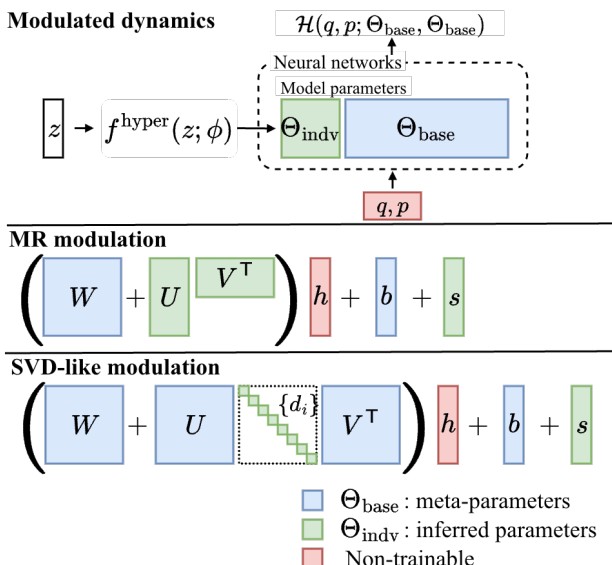

**Modulated dynamics**

**MR modulation**

**SVD-like modulation**

$\boldsymbol{\Theta}_{\text{base}}$ : meta-parameters
$\boldsymbol{\Theta}_{\text{indv}}$ : inferred parameters
Non-trainable

*Figure 2.* Graphical representations of modulated dynamics and layers with two proposed modulation techniques: MR and SVD.

Similarly, SVD-like (abbreviated to SVD) is defined as

$$\boldsymbol{h} \mapsto \sigma\left(\left(\boldsymbol{W}^{(\ell)} + \sum_{i=1}^{r} d_i^{(\ell,k)}\boldsymbol{u}_i^{(\ell)}\boldsymbol{v}_i^{(\ell)\top}\right)\boldsymbol{h} + \boldsymbol{b}^{(\ell)} + \boldsymbol{s}^{(\ell,k)}\right), \tag{5}$$

where

$$\left(\cup_{\ell,i=1}^{L,r}d_i^{(\ell,k)}, \cup_{\ell=1}^{L}\boldsymbol{s}^{(\ell,k)}\right) = \boldsymbol{f}^{\text{hyper}}(\boldsymbol{z}^{(k)}; \boldsymbol{\phi}). \tag{6}$$

Note that the basis vectors $\boldsymbol{u}$ and $\boldsymbol{v}$ are considered as the base parameters $\boldsymbol{\Theta}_{\text{base}}$ in SVD and trained with meta-gradients, while this is not the case for MR. Figure 2 illustrates the base parameters (blue) and the individual parameters (green), inferred from the hypernetwork.

The proposed approaches exploit the local low-rank structure of the parameter space, which ensures that these modulations remain expressive. Specifically, note the following result proven in Appendix D.

**Proposition 3.1.** $P \subset \mathbb{R}^p$ denote a parameter domain and $f : \mathbb{R}^n \times P \to \mathbb{R}^n$ be continuously differentiable at any $\boldsymbol{\mu} \in P$. If the Jacobian $\partial_{\boldsymbol{\mu}}f$ has locally constant rank for $\boldsymbol{x} \in \mathbb{R}^n$ held fixed and $r \leq \min(n, p)$ denotes its maximal rank in $P$, the map $\boldsymbol{\mu} \mapsto \boldsymbol{f}(\boldsymbol{x}, \boldsymbol{\mu})$ has local dimension at most $r$.

This implies that an $r$-dimensional modulation to the parameters of $\boldsymbol{f}$ can be sufficient for capturing local parametric variations no matter the dimension of the parameter space, motivating the low-rank parameterizations introduced here. For SVD-like modulation, we further impose a soft orthogonality constraint on the learned basis matrices $\boldsymbol{U}$ and $\boldsymbol{V}$. This encourages the modulation directions to remain distinct

and non-redundant, mirroring the structure of the classical SVD while not requiring an explicit matrix factorization during training. Specifically, we add Frobenius-norm penalties of the form $\|U^\top U - I\|_F$ and $\|V^\top V - I\|_F$ to the training objective. Moreover, to avoid sign ambiguity in singular values, the hyper-network $f_{\text{hyper}}$ is ReLU-activated.

**Locality regularization** As introduced in (Kirchmeyer et al., 2022), we also enforce a locality constraint restricting instance-specific adaptations of the model away from a shared base representation. Rather than allowing arbitrary deviations when adapting to a new system, the locality constraint limits how far the adapted parameters can move from the global model, effectively biasing learning toward solutions lying in a local neighborhood of the shared parameter space. Accomplishing this amounts to regularizing the magnitude of the latent code and the hyper-network: our implementation applies penalty terms $\lambda_z \|z\|_2$ and $\lambda_\phi \|\phi\|_2$ to all considered modulation techniques.

### 3.3. Meta-learning Algorithm

We now present the meta-learning algorithms employed to train the modulated MLPs described in Appendix A. Recall that each modulated MLP has a set of base model parameters $\Theta_{\text{base}}$ along with individual, instance-specific model parameters $\Theta_{\text{indv}}^{(k)} = z^{(k)}$ captured by the latent codes. In Algorithms 1 and 2, we minimize the velocity matching loss $\|f(\mathcal{T}^{(k)}) - f_{\Theta^{(k)}}(\mathcal{T}^{(k)}; z^{(k)})\|_2^2$ (i.e., Eq. (2) or Eq. (16)) given a set of trajectories denoted by $\mathcal{T}^{(k)} = \{\mathcal{T}^{(k,i)}\}_{i=1}^{n_{\mathcal{T}}}$. This algorithm largely follows the meta-learning strategy presented in the work (Zintgraf et al., 2019; Dupont et al., 2022) and consists essentially of two parts: an inner iteration loop for updating the latent codes (also known as auto-decoding (Park et al., 2019)) and an outer iteration loop for updating the model meta-parameters.

---

**Algorithm 1** Meta-learning algorithm

---

1: **Input:** Training dataset $\mathcal{D}_{\text{train}} = \{\mathcal{T}_{\text{train}}^{(k)}\}_{k=1}^{n_\mu^{\text{train}}}$
2: Initialize the base model parameters $\Theta_{\text{base}}$ and the latent codes $\{z_{\text{train}}^{(k)}\}_{k=1}^{n_\mu^{\text{train}}}$
3: **for** $i_{\text{out}} = 1$ **to** $N_{\text{out}}$ **do**
4:     Sample batch $\mathcal{B}$ of data $\{\mathcal{T}_{\text{train}}^{(b)}\}_{b \in \mathcal{B}}$
5:     **for all** $i_{\text{in}} = 1$ **to** $N_{\text{in}}$ and $b \in \mathcal{B}$ **do**
6:         $z_{\text{train}}^{(b)} \leftarrow z_{\text{train}}^{(b)} - \eta_{\text{in}} \nabla_z \mathcal{L}(\mathcal{T}_{\text{train}}^{(b)})|_{z=z_{\text{train}}^{(b)}}$
7:     **end for**
8:     $\Theta_{\text{base}} \leftarrow \Theta_{\text{base}} - \eta_{\text{out}} \nabla_{\Theta_{\text{base}}} \sum_{b \in \mathcal{B}} \mathcal{L}(\mathcal{T}_{\text{train}}^{(b)})$
9: **end for**

---

Compared to the meta-learning algorithm in (Dupont et al., 2022), we allow the latent codes $z_{\text{train}}^{(k)}$ to evolve over the entire training process instead of resetting them to zero

vectors at every epoch. This means the base parameters $\Theta_{\text{base}}$ are updated with the constantly evolving latent codes during training.

Testing employs a few-shot update (size $N_{\text{test}}$, $N_{\text{test}}$-shot) of the latent code $z^{(b)}$ using data collected from a target system in the test set. The base model parameters are fixed during this process, reflecting auto-decoding on the test dataset. We initialize the test latent codes $\{z_{\text{test}}^{(b)}\}$ by assigning $z_{\text{test}}^{(b)} = z_{\text{avg}}, \forall b \in \mathcal{B}$, in terms of the Euclidean mean of the training latent codes: $z_{\text{avg}} = \frac{1}{n_\mu^{\text{train}}} \sum_{k=1}^{n_\mu^{\text{train}}} z_{\text{train}}^{(k)}$. Empirically, we show that this strategy of evolving latent codes during training and initializing test codes with $z_{\text{avg}}$ consistently improves model performance in the considered scenarios. It is also shown in Appendix D that the proposed approach retains the structure-preserving properties of the Hamiltonian and GENERIC modeling strategies.

---

**Algorithm 2** Test algorithm

---

1: Set $\mathcal{B} = \mathcal{D}_{\text{test}}$ and initialize $\{z_{\text{test}}^{(b)}\}_{b \in \mathcal{B}}$
2: **for all** $i_{\text{test}} = 1$ **to** $N_{\text{test}}$ and $b \in \mathcal{B}$ **do**
3:     $z_{\text{test}}^{(b)} \leftarrow z_{\text{test}}^{(b)} - \eta_{\text{test}} \nabla_z \mathcal{L}(\mathcal{T}_{\text{test}}^{(b)})|_{z=z_{\text{test}}^{(b)}}$
4: **end for**

---

## 4. Experimental Setup

The next goal is to compare the proposed modulation-based meta-learning strategies to each other and to popular optimization-based strategies including MAML (Finn et al., 2017), Reptile (a first-order MAML) (Nichol et al., 2018), and almost no inner loop (ANIL) (Raghu et al., 2020). For modulation-based meta-learning, we consider the latent FW modulation technique, adapted from (Kirchmeyer et al., 2022) and also known as CODA, which is—to our knowledge—the only modulation-based method previously studied in the context of dynamics learning. We additionally include the latent shift modulation method (Dupont et al., 2022) (Shift) as a lightweight modulation baseline. The FW method applies layer-wise corrections to both weights and biases of the base model, while Shift applies only layer-wise bias modulations. We consider two variants for each of the proposed latent MR and SVD modulation strategies. These are denoted MR(5) and RO (=MR(1)) in the MR case, reflecting rank-5 and rank-1 adaptation, respectively. Similarly, SVD and SVD(5) are used to denote full-rank and rank-5 adaptation, respectively, in the SVD case. All strategies are compared to standard training, denoted "Scratch", which trains an individual HNN (Greydanus et al., 2019) or GNN (Lee et al., 2021a) for each parameter instance $\mu$, and all methods are trained by minimizing either the symplecticity loss (Eq. (2)) in the Hamiltonian case, or the GENERIC loss (Eq. (16)) in the dissipative case.

**Datasets** The approaches are compared using standard benchmark problems in the literature. We consider 3 Hamiltonian systems: the mass-spring system, ideal pendulum, and Duffing oscillator. In the dissipative case, we consider a damped nonlinear oscillator (DNO) (Shang & Öttinger, 2020).

**Data generation** For each system considered, we generate 80 Hamiltonian/GENERIC functions by randomly selecting parameter instances $\boldsymbol{\mu}^{(k)}$, $k = 1, \ldots, 80$, splitting them into 70 and 10 for train and test datasets, respectively. For the $k^{\text{th}}$ such function, we generate a set of $n_{\mathcal{T}} = 10$ trajectories $\mathcal{T}^{(k)}$ using random initial conditions. At test time, these trajectories are used to update the latent codes via Algorithm 2. For evaluation only, an additional $n_{\mathcal{T}}$ trajectories $\mathcal{T}^{(k)}_{\text{perf}}$ are sampled from novel initial conditions and held out for measuring the model's performance. For all systems, ground-truth trajectories $(\boldsymbol{q}, \boldsymbol{p})$ are computed via numerical simulations. Derivatives $\dot{\boldsymbol{q}}, \dot{\boldsymbol{p}}$ are computed using centered finite differences. See the Appendix for details.

The default setup for the neural network defining the dynamics function is an MLP with 2 layers of 100 neurons and sigmoid linear unit (SiLU) activation (Hendrycks, 2016). The hyper-network is set as a single layer MLP with no nonlinearity i.e., an affine embedding $\boldsymbol{f}_{\text{hyper}}(\boldsymbol{z}^{(k)}; \boldsymbol{\phi}) = \boldsymbol{W}\boldsymbol{z}^{(k)} + \boldsymbol{b}$ for $\boldsymbol{\phi} = \{\boldsymbol{W}, \boldsymbol{b}\}$. The size of the weight matrix, $\boldsymbol{W}$, is determined by the latent modulation vector, $\boldsymbol{z}^{(k)}$, and the corresponding output, $\boldsymbol{\Theta}^{(k)}$.

**Evaluation metric** To evaluate the learned models, we employ several metrics that assess different aspects of their performance. The first error metric is relative $\ell^2$-error applied to the model's output, measured using the test trajectories, $\mathcal{T}^{(k)}_{\text{perf}}$. This error is defined as $\epsilon_{\text{traj}} = \frac{1}{10 n_{\mathcal{T}}} \sum_{k=1}^{10} \sum_{i=1}^{n_{\mathcal{T}}} \frac{\|\boldsymbol{f}(\mathcal{T}^{(k,i)}; \boldsymbol{\mu}^{(k)}) - \boldsymbol{f}_{\Theta}(\mathcal{T}^{(k,i)}; \boldsymbol{\mu}^{(k)})\|_2}{\|\boldsymbol{f}(\mathcal{T}^{(k,i)}; \boldsymbol{\mu}^{(k)})\|_2}$, i.e., the relative error, measured over $n_{\mathcal{T}}$ trajectories for each test system, averaged over 10 systems. The second error metric is the same relative $\ell^2$ error but measured on a uniform mesh grid $\Omega_h$. This error is defined as $\epsilon_{\text{field}} = \frac{1}{10} \sum_{k=1}^{10} \frac{\|\boldsymbol{f}(\Omega_h; \boldsymbol{\mu}^{(k)}) - \boldsymbol{f}_{\Theta}(\Omega_h; \boldsymbol{\mu}^{(k)})\|_2}{\|\boldsymbol{f}(\Omega_h; \boldsymbol{\mu}^{(k)})\|_2}$, i.e., the averaged relative error measured on a uniform grid. This performance metric serves as an indicator of how well the models perform on out-of-distribution samples.

## 5. Results

In this section, we provide results comparing the performance of the proposed methods with the baselines.

### 5.1. Summary Results

Table 1 provides information on the performance of all considered models measured by the field errors $\epsilon_{\text{field}}$ and the trajectory errors $\epsilon_{\text{traj}}$. Overall, the optimization-based algorithms demonstrate potential for improving performance over standard training algorithms for HNNs. For example, Reptile for Duffing/mass-spring and ANIL for mass spring/pendulum are significantly improved over training from scratch. However, all modulation-based algorithms improve the performance by about 65% compared to optimization-based methods, demonstrating the strong performance of modulation-based meta-learning in practice. Among these techniques, SVD-like decompositions (in particular, SVD(5)) achieve the lowest errors across all benchmark problems.

*Table 1.* [Energy-conserving systems] Model performance measured in the field errors, $\epsilon_{\text{field}}$ ($\downarrow$), and the trajectory errors, $\epsilon_{\text{traj}}$ ($\downarrow$), (lower values indicate better performance). The numbers indicate the mean ($\times 10^{-2}$) and the standard deviation (inside the parenthesis, $\times 10^{-3}$), obtained over 5 individual runs.

| | Duffing | | Mass Spring | | Pendulum | |
|---|---|---|---|---|---|---|
| | $\epsilon_{\text{field}}$ | $\epsilon_{\text{traj}}$ | $\epsilon_{\text{field}}$ | $\epsilon_{\text{traj}}$ | $\epsilon_{\text{field}}$ | $\epsilon_{\text{traj}}$ |
| Scratch | 41.84 (4.08) | 45.77 (8.35) | 58.40 (15.5) | 39.45 (9.44) | 83.35 (10.7) | 79.84 (8.62) |
| MAML | 92.90 (18.1) | 20.33 (90.8) | 320.5 (186) | 30.00 (82.6) | 99.13 (98.8) | 52.37 (37.6) |
| Reptile | 18.55 (11.1) | 21.76 (29.8) | 43.27 (5.43) | 39.36 (51.2) | 88.72 (26.3) | 75.73 (18.5) |
| ANIL | 94.01 (5.23) | 41.24 (46.8) | 30.76 (9.28) | 25.70 (4.67) | 35.07 (10.5) | 37.33 (41.7) |
| FW | 10.30 (2.06) | 2.784 (1.70) | 1.600 (1.62) | 1.307 (2.86) | 8.231 (1.84) | 10.65 (2.70) |
| Shift | 9.031 (2.15) | 5.952 (5.48) | 16.66 (16.6) | 13.18 (17.7) | 9.758 (9.77) | 12.88 (9.81) |
| MR | 10.15 (0.69) | 2.395 (3.36) | 1.853 (9.83) | 1.387 (2.25) | 7.069 (7.37) | 8.323 (9.59) |
| RO | 10.10 (0.25) | 2.593 (2.32) | 1.517 (1.21) | 1.370 (2.10) | 6.473 (7.52) | 8.271 (14.1) |
| SVD | 10.18 (1.80) | 2.328 (1.80) | 1.511 (2.72) | 1.196 (3.26) | 5.088 (6.62) | 6.726 (12.4) |
| SVD(5) | 10.03 (0.55) | 2.303 (1.84) | 1.509 (4.14) | 1.116 (3.34) | 4.624 (4.80) | 5.336 (5.06) |

**Structural similarity** Considering the learned vector field as an image depicted on 2d space, we compute the structural similarity (SSIM) (Wang et al., 2004) between $\boldsymbol{f}(\Omega_h; \boldsymbol{\mu}^{(k)})$ and $\boldsymbol{f}_{\Theta}(\Omega_h; \boldsymbol{\mu}^{(k)})$. This measure takes the magnitude and direction of output vectors into account, computing similarities in terms of intensity and patterns. As reported in Table 2, the results of measuring structural similarity via SSIM further support that the modulation-based methods demonstrate improved capability in capturing the target vector fields compared to optimization-based meta-learning algorithms. Among the modulation-based methods, the SVD-like modulation approaches outperform all other baselines.

### 5.2. Detailed Analysis of the Pendulum

To further analyze the results on energy-conserving systems, the pendulum is presented here in detail. Due to space constraints, readers are referred to the Appendix for similarly detailed results on other systems.

**Parameter efficiency** To assess parameter efficiency, we measure the size of the hyper-network, which dominates model size in modulation-based meta-learning. For a fixed base architecture, this directly reflects how efficiently a modulation strategy adapts to new system instances. We

*Table 2.* [Energy-conserving systems] Model performance measured in SSIM ($\uparrow$). The numbers indicate the mean and the standard deviation (inside the parenthesis, $\times 10^{-3}$), obtained over 5 individual runs.

|         | Duffing       | Mass Spring   | Pendulum      |
|---------|---------------|---------------|---------------|
| Scratch | 0.606 (7.44)  | 0.397 (18.3)  | 0.350 (11.8)  |
| MAML    | 0.521 (47.9)  | 0.203 (18.0)  | 0.374 (23.1)  |
| Reptile | 0.927 (9.12)  | 0.555 (23.3)  | 0.242 (32.7)  |
| ANIL    | 0.556 (6.13)  | 0.796 (14.3)  | 0.778 (12.0)  |
| FW      | 0.994 (0.25)  | 0.996 (0.70)  | 0.964 (10.9)  |
| Shift   | 0.993 (0.49)  | 0.959 (3.41)  | 0.959 (5.35)  |
| MR(5)   | 0.995 (0.14)  | 0.995 (0.94)  | 0.971 (6.71)  |
| RO      | 0.995 (0.15)  | 0.997 (1.30)  | 0.981 (4.86)  |
| SVD     | 0.995 (0.18)  | 0.997 (0.90)  | 0.988 (3.45)  |
| SVD(5)  | 0.995 (0.11)  | 0.998 (1.26)  | 0.989 (1.43)  |

evaluate the trade-off between performance and parameter cost by plotting model accuracy against the relative hyper-network size on the pendulum system. Using the shift-modulation hyper-network (the smallest among all methods) as a reference, the sizes of other models are reported as multiplicative factors relative to this baseline.

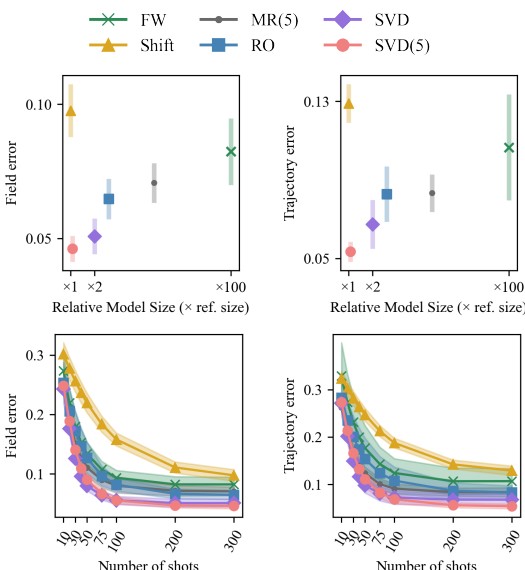

*Figure 3.* [Pendulum] Field errors and trajectory errors over the number of shots in the $n$-shot update during the test phase. (Top) the horizontal axis in log-scale.

Figure 3 reports these results. While FW achieves strong performance, it requires substantially larger hyper-networks. In contrast, the SVD modulation achieves superior performance with significantly fewer additional parameters. Moreover, the bottom panels show field and trajectory errors as functions of the number of shots used for latent-code adaptation at test time. SVD-like modulation consistently achieves lower errors across all shot counts, indicating superior few-

shot efficiency.

**Phase-space visualization** Figure 4 shows phase-space $(q, p)$ vector fields for a representative pendulum system at a fixed parameter $\boldsymbol{\mu}^{(k)}$, including the ground-truth field and the fields obtained after 300-shot adaptation using all methods. In the first panel, red trajectories from $\mathcal{T}_{\text{test}}^{(k)}$ are used for auto-decoding (Algorithm 2), while green trajectories are reserved for evaluating the trajectory error $\epsilon_{\text{traj}}$.

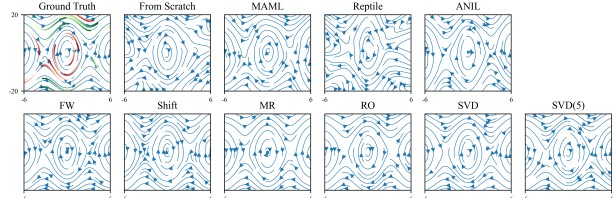

*Figure 4.* [Pendulum] Phase space vector fields obtained through 300-shot adaptation to target fields during the test phase.

**Locality constraints** We evaluate the effect of the locality constraint by varying the regularization weights $\lambda_\phi \in \{10^{-2}, 10^{-3}, 10^{-4}\}$ and $\lambda_z \in \{10^{-1}, 10^{-2}, 10^{-3}\}$. For each $(\lambda_\phi, \lambda_z)$ pair, five independent runs are performed and results are averaged across all settings, capturing both mean performance and variance. Across all methods, SVD-based modulation achieves consistently strong performance with lower variance, indicating robust adaptation across a wide range of locality strengths.

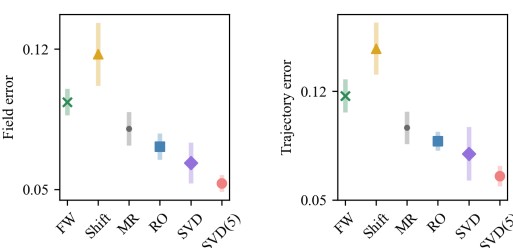

*Figure 5.* [Pendulum] Performance averaged across varying values of $\lambda_\phi$ and $\lambda_z$.

**Initialization schemes** Figure 5 compares two latent vector initialization schemes described in Section 3.3: zero initialization (crosses) and the proposed initialization (Circles) using the averaged latent codes obtained during training. Across all settings, the proposed method consistently yields lower errors, demonstrating more effective and stable test-time adaptation. The observed gap between zero and mean initialization suggests that the learned latent distribution is not centered at the origin. Mean initialization therefore provides better alignment with the learned latent space and improves adaptation performance. Incorporating a centered

prior during training may reduce this gap, which we leave for future investigation.

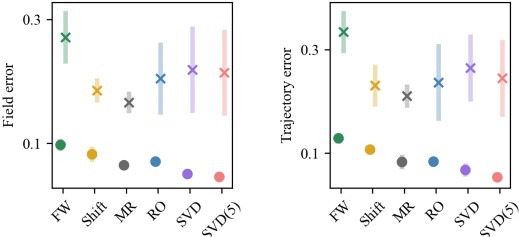

*Figure 6.* [Pendulum] Performance comparisons between the proposed and zero latent initialization schemes. The markers ● and × indicate the proposed and zero initialization, respectively.

### 5.3. Multi-domain Evaluation

To evaluate multi-domain behavior, we train a single base network using a combined dataset consisting of multiple dynamical systems, namely the Duffing oscillator, mass-spring, and pendulum. In contrast to previous experiments where a separate base model is trained per system, this setting tests whether modulation alone provides sufficient expressivity to adapt across heterogeneous dynamics. During training, trajectories from all three systems are jointly used, while system-specific adaptation is handled exclusively through latent modulation.

Figure 7 reports the resulting field and trajectory errors for each method (again, showing only the results of the case when the target system in the test phase is pendulum). Among all approaches, SVD-like modulations consistently achieve the best performance, demonstrating superior expressivity and robustness in multi-domain settings.

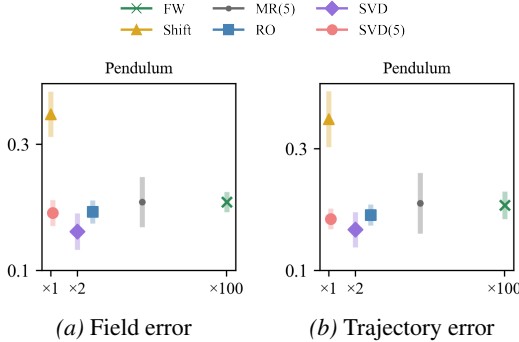

*(a)* Field error       *(b)* Trajectory error

*Figure 7.* [Multi-domain] Field errors and trajectory errors

### 5.4. Dissipative System

The DNO is presented here to illustrate the results of our proposed modulation approaches on structure-preserving approaches to dissipative systems. The optimization methods, Reptile and ANIL, struggle to achieve stable training,

either producing Not-a-Number (NaN) or diverging (Div.) results. Modulation-based approaches (except MR) outperform optimization-based methods, with RO and SVD(3) achieving the best performance; SVD(3) is the most robust across five random seeds.

*Table 3.* [Energy dissipative system, DNO] Model performance measured in the trajectory errors, $\epsilon_{\text{traj}}$ ($\downarrow$) and the field errors, $\epsilon_{\text{field}}$ ($\downarrow$) (lower values indicate better performance). The numbers indicate the mean ($\times 10^{-1}$) and the standard deviation (inside the parenthesis, $\times 10^{-1}$), obtained over 5 individual runs.

|  | $\epsilon_{\text{field}}$ | $\epsilon_{\text{traj}}$ |
|---|---|---|
| Scratch | 5.830 (10.3) | 8.421 (28.2) |
| MAML | 3.194 (21.5) | 4.055 (21.7) |
| Reptile | NaN | NaN |
| ANIL | Div. | Div. |
| FW | 2.091 (40.9) | 1.419 (24.3) |
| Shift | 1.158 (5.23) | 1.668 (6.77) |
| MR | 7.880 (1.88) | 42.50 (6.79) |
| RO | .9565 (11.1) | 1.155 (12.5) |
| SVD | 1.221 (1.48) | 1.357 (1.51) |
| SVD(3) | 1.025 (.567) | 1.423 (.801) |

## 6. Related Work

**Structure-preserving neural networks** Designing neural network architectures that exactly enforce important physical properties has been an important and extensively studied topic. Parameterization techniques that preserve physical structure include HNNs (Greydanus et al., 2019; Toth et al., 2020), Lagrangian neural networks (Cranmer et al., 2020; Lutter et al., 2018), port-Hamiltonian neural networks (Zhong et al., 2020; Desai et al., 2021), and GENERIC/metriplectic neural networks (Hernandez et al., 2021; Lee et al., 2021a; Zhang et al., 2022; Lee et al., 2022; Gruber et al., 2023; 2025). Neural network architectures that mimic the action of symplectic integrators have been proposed in (Chen et al., 2019; Jin et al., 2020; Tong et al., 2021). Extensions of HNNs to noncanonical Lie–Poisson systems and generalized coordinates have been studied in (Finzi et al., 2020; Chen et al., 2021; Eldred et al., 2024).

**Meta-learning structure-preserving systems** A handful of studies have investigated meta-learning algorithms for structure-preserving systems, all focused on energy-conserving dynamics (Lee et al., 2021b; Iwata & Tanaka, 2024; Song & Jeong, 2024); in (Lee et al., 2021b) and (Song & Jeong, 2024), optimization-based meta-learning algorithms have been explored (i.e., MAML and ANIL). In (Song & Jeong, 2024), a generalization between different Hamiltonian systems has been studied. In (Iwata & Tanaka, 2024), Gaussian processes-based meta-learning has been studied. In contrast, meta-learning for dissipative dynamics has received little attention. Although dissipative systems were considered in (Iwata & Tanaka, 2024; Song & Jeong,

2024), these works did not consider structure-preserving techniques that satisfy thermodynamical laws.

**Meta-learning neural representations**  For NeRFs and INRs, an optimization-based meta learning algorithm, MAML (Finn et al., 2017), has been the popular choice for meta-learning. In (Tancik et al., 2021; Sitzmann et al., 2020), the initialization of all model parameters is meta-learned through meta-gradient updates and then fit to each data point. In (Dupont et al., 2022), a different meta-learning algorithm leveraging a latent (i.e., modulation) vector storing individual information is proposed, resembling CAVIA-type fast adaptation methods (Zintgraf et al., 2019).

## 7. Conclusion

This study has explored meta-learning algorithms for learning structure-preserving dynamics in many-query or parameter-varying scenarios. To address the limitations of optimization-based meta-learning approaches, we introduced modulation-based techniques that adapt model parameters to novel system configurations by conditioning on a low-dimensional latent vector. Specifically, two novel modulation strategies were proposed which offer greater expressivity compared to existing methods: latent multi-rank and latent SVD-like modulation. These were integrated into a meta-learning framework that updates model parameters effectively through auto-decoding, enabling a method for the structure-preserving learning of system dynamics that does not depend on explicit knowledge of its parametric dependence. This approach was evaluated on three energy-conserving and one dissipative system, demonstrating that the proposed methods consistently outperform baseline approaches across several relevant performance metrics.

## Impact Statement

This work contributes to machine learning for science by advancing meta-learning methods for structure-preserving dynamical systems. Many physical phenomena are governed by energy-conserving and dissipative formalisms and are commonly encountered in many-query settings in practice. The methods developed here may provide a scalable foundation for data-driven modeling of real-world physical systems, with potential relevance to domains such as thermodynamics and molecular dynamics. We do not anticipate direct ethical or societal concerns arising from this work, which is primarily methodological in nature.

## Acknowledgements

K.L. acknowledges support from the U.S. National Science Foundation under grant IIS 2338909. A.G. acknowledges support from the U.S. Department of Energy, Office of Advanced Scientific Computing Research under the Scalable, Efficient, and Accelerated Causal Reasoning for Earth and Embedded Systems (SEA-CROGS) project. Sandia National Laboratories is a multimission laboratory managed and operated by National Technology & Engineering Solutions of Sandia, LLC, a wholly owned subsidiary of Honeywell International Inc., for the U.S. Department of Energy's National Nuclear Security Administration under contract DE-NA0003525. This paper describes objective technical results and analysis. Any subjective views or opinions that might be expressed in the paper do not necessarily represent the views of the U.S. Department of Energy or the United States Government. This article has been co-authored by an employee of National Technology & Engineering Solutions of Sandia, LLC under Contract No. DE-NA0003525 with the U.S. Department of Energy (DOE). The employee owns all right, title and interest in and to the article and is solely responsible for its contents. The United States Government retains and the publisher, by accepting the article for publication, acknowledges that the United States Government retains a non-exclusive, paid-up, irrevocable, world-wide license to publish or reproduce the published form of this article or allow others to do so, for United States Government purposes. The DOE will provide public access to these results of federally sponsored research in accordance with the DOE Public Access Plan https://www.energy.gov/downloads/doe-public-access-plan.

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

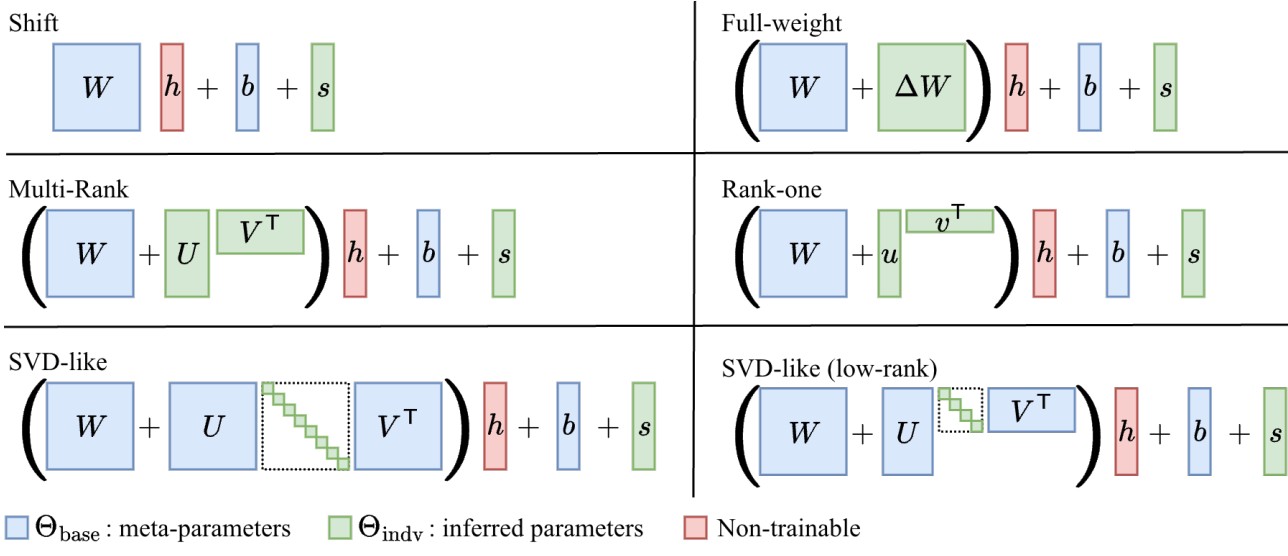

*Figure 8.* Graphical representations of layers with all considered modulation techniques.

## A. Modulation Techniques

For an MLP with weights and biases $\{\boldsymbol{W}^{(\ell)}, \boldsymbol{b}^{(\ell)}\}_{\ell=1}^{L}$, we now define various modulation techniques utilized in our study. For all techniques, we explicitly write an expression for the $k$-th Hamiltonian function and its corresponding $k$-th latent code: $\mathcal{H}^{(k)}$ and $\boldsymbol{z}^{(k)}$.

The first two modulations considered are existing methods; the latent shift modulation proposed in (Dupont et al., 2022) and the latent full-weight modulation (Kirchmeyer et al., 2022).[1] The latent shift modulation technique can be defined as the mapping:

$$\boldsymbol{h} \mapsto \sigma\left(\boldsymbol{W}^{(\ell)}\boldsymbol{h} + \boldsymbol{b}^{(\ell)} + \boldsymbol{s}^{(\ell,k)}\right), \tag{7}$$

where $\boldsymbol{s}^{(k)} = \cup_{\ell=1}^{L}\boldsymbol{s}^{(\ell,k)} = \boldsymbol{f}_{\text{hyper}}(\boldsymbol{z}^{(k)}; \boldsymbol{\phi})$.

The latent full weight modulation can be similarly defined as:

$$\boldsymbol{h} \mapsto \sigma\left((\boldsymbol{W}^{(\ell)} + \Delta\boldsymbol{W}^{(\ell,k)})\boldsymbol{h} + \boldsymbol{b}^{(\ell)} + \boldsymbol{s}^{(\ell,k)}\right), \tag{8}$$

where $(\Delta\boldsymbol{W}^{(k)}, \boldsymbol{s}^{(k)}) = \left(\cup_{\ell=1}^{L}\Delta\boldsymbol{W}^{(\ell,k)}, \cup_{\ell=1}^{L}\boldsymbol{s}^{(\ell,k)}\right) = \boldsymbol{f}_{\text{hyper}}(\boldsymbol{z}^{(k)}; \boldsymbol{\phi})$.

The two proposed modulation techniques discussed in Section 3 are reiterated here. The multi-rank modulation technique is defined as:

$$\boldsymbol{h} \mapsto \sigma\left((\boldsymbol{W}^{(\ell)} + \boldsymbol{U}^{(\ell,k)}\boldsymbol{V}^{(\ell,k)\mathsf{T}})\boldsymbol{h} + \boldsymbol{b}^{(\ell)} + \boldsymbol{s}^{(\ell,k)}\right), \tag{9}$$

where $(\boldsymbol{U}^{(k)}, \boldsymbol{V}^{(k)}, \boldsymbol{s}^{(k)}) = \left(\cup_{\ell=1}^{L}\boldsymbol{U}^{(\ell,k)}, \cup_{\ell=1}^{L}\boldsymbol{s}^{(\ell,k)}, \cup_{\ell=1}^{L}\boldsymbol{V}^{(\ell,k)}\right) = \boldsymbol{f}_{\text{hyper}}(\boldsymbol{z}^{(k)}; \boldsymbol{\phi})$.

Finally, the SVD-like modulation technique is defined as:

$$\boldsymbol{h} \mapsto \sigma\left((\boldsymbol{W}^{(\ell)} + \sum_{i=1}^{r} d_i^{(\ell,k)}\boldsymbol{u}_i^{(\ell)}\boldsymbol{v}_i^{(\ell)\mathsf{T}})\boldsymbol{h} + \boldsymbol{b}^{(\ell)} + \boldsymbol{s}^{(\ell,k)}\right), \tag{10}$$

where $(\boldsymbol{d}^{(k)}, \boldsymbol{s}^{(k)}) = \left(\cup_{\ell=1}^{L} d^{(\ell,k)}, \cup_{\ell=1}^{L}\boldsymbol{s}^{(\ell,k)}\right) = \boldsymbol{f}_{\text{hyper}}(\boldsymbol{z}^{(k)}; \boldsymbol{\phi})$.

---

[1]Note that we do not consider the advanced hypernetwork architectures or other implementation tricks studied in previous works. Instead, using the same hypernetwork architecture for every modulation technique enables us to focus on the core structure of the proposed modulations themselves.

**Hyper-network parameter scaling** The parameter cost of the hyper-network scales differently across modulation strategies as a function of the network depth $l$, width $w$, and rank $r$. FW generates both weights and biases for each layer, resulting in a quadratic scaling of $O(lw^2 + lw)$. Shift modulation produces only layer-wise bias terms and therefore scales linearly as $O(lw)$. Multi-rank (MR) modulation outputs two low-rank factor matrices per layer along with biases, yielding a scaling of $O(2lrw + lw)$, which reduces to $O(3lw)$ when $r = 1$. In contrast, SVD-like modulation is significantly more parameter efficient: the full SVD-like variant scales as $O(2lw)$, while the low-rank SVD($r$) formulation scales as $O(lr + lw)$. These scalings highlight that SVD-like modulation provides a favorable balance between expressivity and hyper-network parameter cost.

*Table 4.* Asymptotic scaling of hyper-network parameters for different modulation strategies.

| Method | Hyper-network scaling |
|---|---:|
| Shift | $O(lw)$ |
| FW | $O(lw^2 + lw)$ |
| MR ($r$) | $O(2lrw + lw)$ |
| RO | $O(3lw)$ |
| SVD (full) | $O(2lw)$ |
| SVD ($r$) | $O(lr + lw)$ |

## B. GENERIC Formalism

Within the GENERIC formalism, the time evolution of an observable $A(\boldsymbol{x})$ is governed by a combination of reversible and irreversible dynamics:

$$\frac{\mathrm{d}A}{\mathrm{d}t} = \{A, E\} + [A, S], \tag{11}$$

where $E$ and $S$ represent the generalized free energy and entropy, respectively. The binary operations $\{\cdot, \cdot\}$ and $[\cdot, \cdot]$ correspond to the Poisson bracket and the dissipative bracket of the system, defined in terms of a skew-symmetric Poisson matrix $\boldsymbol{L} = \boldsymbol{L}(\boldsymbol{x})$ resp. a symmetric positive semi-definite (SPSD) friction matrix $\boldsymbol{M} = \boldsymbol{M}(\boldsymbol{x})$ generally depending on the state $\boldsymbol{x}$—that is,

$$\{A, B\} = \frac{\partial A}{\partial \boldsymbol{x}}^{\mathsf{T}} \boldsymbol{L}(\boldsymbol{x}) \frac{\partial B}{\partial \boldsymbol{x}}, \quad \text{and} \quad [A, B] = \frac{\partial A}{\partial \boldsymbol{x}}^{\mathsf{T}} \boldsymbol{M}(\boldsymbol{x}) \frac{\partial B}{\partial \boldsymbol{x}}. \tag{12}$$

To ensure thermodynamic consistency, the GENERIC framework imposes two degeneracy conditions on the kernels of $\boldsymbol{L}$ and $\boldsymbol{M}$:

$$\boldsymbol{L}(\boldsymbol{x}) \frac{\partial S}{\partial \boldsymbol{x}} = \boldsymbol{0}, \quad \text{and} \quad \boldsymbol{M}(\boldsymbol{x}) \frac{\partial E}{\partial \boldsymbol{x}} = \boldsymbol{0}. \tag{13}$$

With these degeneracy conditions, the first and the second laws of thermodynamics—energy conservation and entropy generation—are satisfied, as seen through the rates of change of $E, S$,

$$\begin{aligned}
\frac{\mathrm{d}E}{\mathrm{d}t} &= \{E, E\} + [E, S] \overset{①}{=} \{E, E\} \overset{②}{=} 0, \quad \text{and} \\
\frac{\mathrm{d}S}{\mathrm{d}t} &= \{S, E\} + [S, S] \overset{①}{=} [S, S] \overset{②}{\geq} 0,
\end{aligned} \tag{14}$$

where the equalities ① are due to the degeneracy conditions and the equalities ② are due to the symmetry and definiteness properties of $\boldsymbol{L}$ and $\boldsymbol{M}$, respectively. For state variables $\boldsymbol{x} = [\boldsymbol{q}, \boldsymbol{p}, S]^{\mathsf{T}}$, the full dynamics are given by

$$\frac{\mathrm{d}\boldsymbol{x}}{\mathrm{d}t} = \boldsymbol{L}(\boldsymbol{x}) \frac{\partial E}{\partial \boldsymbol{x}} + \boldsymbol{M}(\boldsymbol{x}) \frac{\partial S}{\partial \boldsymbol{x}}. \tag{15}$$

Various techniques have been developed to parameterize metriplectic dynamics in a way that exactly preserves the desired structural properties Eq. (13) (Lee et al., 2021a; 2022; Gruber et al., 2023; 2025). Such approaches are collectively referred to as GENERIC neural networks (GNNs) in this work. Essentially, the four component fields $\boldsymbol{L}, \boldsymbol{M}, E$, and $S$ are parameterized as neural networks or trainable arrays with prescribed mathematical structure, $\boldsymbol{L}_{\Theta_L}(\boldsymbol{x}), \boldsymbol{M}_{\Theta_M}(\boldsymbol{x}), E_{\Theta_E}(\boldsymbol{x})$, and $S_{\Theta_S}(\boldsymbol{x})$, respectively. By explicitly accounting for operator symmetries, definiteness properties, and degeneracy conditions Eq. (13) in the network parameterizations, metriplectic structure in the dynamics is preserved by construction, independent of training

quality; we refer readers to Appendix for the detailed descriptions of existing parameterization techniques. The resulting models can be trained from data for the total state derivative $\frac{d\boldsymbol{x}}{dt}$ by minimizing a loss analogous to the symplecticity loss from before,

$$\mathcal{L}_{\text{generic}} = \left\| \frac{d\boldsymbol{x}}{dt} - \boldsymbol{L}_{\Theta_L}(\boldsymbol{x})\frac{\partial E_{\Theta_E}}{\partial \boldsymbol{x}} - \boldsymbol{M}_{\Theta_M}(\boldsymbol{x})\frac{\partial S_{\Theta_S}}{\partial \boldsymbol{x}} \right\|_p^2. \tag{16}$$

**Remark.** For canonical coordinates $\boldsymbol{x} = (\boldsymbol{q}, \boldsymbol{p}, S)$, and the canonical Poisson matrix $\boldsymbol{L} = [[\boldsymbol{0}; \boldsymbol{I}; 0], [-\boldsymbol{I}; \boldsymbol{0}; 0], [0; 0; 0]]$, and $\boldsymbol{M} = 0$, GENERIC dynamics recovers Hamiltonian dynamics in the variables $(\boldsymbol{q}, \boldsymbol{p})$. It has been empirically shown that training GNNs on energy-conserving trajectories can result in energy conserving dynamics (i.e., $\boldsymbol{M} = 0$) while HNNs fail to capture dissipative dynamics in the previous work (Lee et al., 2021a; Gruber et al., 2025).

## C. GENERIC Neural Networks

### C.1. Parameterization Techniques in (Lee et al., 2021a)

As described in Section 2.2, the GENERIC formalism represents the system dynamics as

$$\frac{d\boldsymbol{x}}{dt} = \boldsymbol{L}\frac{\partial E}{\partial \boldsymbol{x}} + \boldsymbol{M}\frac{\partial S}{\partial \boldsymbol{x}}, \tag{17}$$

where $\boldsymbol{L} = \boldsymbol{L}(\boldsymbol{x})$ denotes a skew-symmetric Poisson matrix and $\boldsymbol{M} = \boldsymbol{M}(\boldsymbol{x})$ denotes a symmetric positive semi-definite (SPSD) friction matrix. The two degeneracy conditions that must be satisfied to ensure thermodynamic consistency are given by

$$\boldsymbol{L}\frac{\partial S}{\partial \boldsymbol{x}} = \boldsymbol{0}, \quad \text{and} \quad \boldsymbol{M}\frac{\partial E}{\partial \boldsymbol{x}} = \boldsymbol{0}. \tag{18}$$

In this work, we consider a specific parameterization of GENERIC inherited from (Öttinger, 2014; Lee et al., 2021a) due to their simplicity. Its basic form and the necessary modifications to enable meta-learning are presented in the following.

We first describe the original formulation, followed by the parts that are modified to make it suitable for meta-learning.

In the original work (Lee et al., 2021a), the energy and entropy are represented as neural networks, i.e., $E(\boldsymbol{x}) \approx E_{\Theta_E}(\boldsymbol{x})$ and $S(\boldsymbol{x}) \approx S_{\Theta_S}(\boldsymbol{x})$, where $\Theta_E$ and $\Theta_S$ denote weights and biases of the networks corresponding to $E$ and $S$, respectively. The reversible part of the dynamics is then characterized by a skew-symmetric and state-dependent Poisson bracket,

$$\{A, B\} = \boldsymbol{\xi}_{\alpha\beta\gamma}\frac{\partial A}{\partial \boldsymbol{x}_\alpha}\frac{\partial B}{\partial \boldsymbol{x}_\beta}\frac{\partial S}{\partial \boldsymbol{x}_\gamma}, \tag{19}$$

where $\boldsymbol{\xi}_{\alpha\beta\gamma}$ is a totally anti-symmetric 3d tensor without state dependence. To enforce this anti-symmetry exactly, the authors consider a generic 3 tensor $\tilde{\boldsymbol{\xi}}_{\alpha\beta\gamma}$ with learnable entries and apply the following skew-symmetrization trick,

$$\boldsymbol{\xi}_{\alpha\beta\gamma} = \frac{1}{3!}\left( \tilde{\boldsymbol{\xi}}_{\alpha\beta\gamma} - \tilde{\boldsymbol{\xi}}_{\alpha\gamma\beta} + \tilde{\boldsymbol{\xi}}_{\beta\gamma\alpha} - \tilde{\boldsymbol{\xi}}_{\beta\alpha\gamma} + \tilde{\boldsymbol{\xi}}_{\gamma\alpha\beta} - \tilde{\boldsymbol{\xi}}_{\gamma\beta\alpha} \right). \tag{20}$$

With this total antisymmetry, one can easily verify that $\{E, E\} = 0$ and $\{S, E\} = 0$. Moreover, the corresponding $\boldsymbol{L}$ matrix is now extracted through contraction against the last index, i.e., $\boldsymbol{L}(\boldsymbol{x}) = \boldsymbol{\xi}_{\alpha\beta\gamma}\frac{\partial S}{\partial \boldsymbol{x}_\gamma}$ assuming the Einstein summation convention.

Similarly, the irreversible part of the dynamics is characterized by an SPSD irreversible bracket,

$$[A, B] = \boldsymbol{\zeta}_{\alpha\beta,\mu\nu}\frac{\partial A}{\partial \boldsymbol{x}_\alpha}\frac{\partial E}{\partial \boldsymbol{x}_\beta}\frac{\partial B}{\partial \boldsymbol{x}_\mu}\frac{\partial E}{\partial \boldsymbol{x}_\nu}, \tag{21}$$

where the 4d tensor $\boldsymbol{\zeta}$ is defined as

$$\boldsymbol{\zeta}_{\alpha\beta,\mu\nu} = \boldsymbol{\Lambda}_{\alpha\beta}^m \boldsymbol{D}_{mn} \boldsymbol{\Lambda}_{\mu\nu}^n. \tag{22}$$

Here, $\boldsymbol{\Lambda}$ is a 3d tensor skew-symmetric in its lower indices, and $\boldsymbol{D}_{mn}$ is an SPSD matrix, meaning that

$$\boldsymbol{\Lambda}_{\alpha\beta}^m = -\boldsymbol{\Lambda}_{\beta\alpha}^m, \quad \text{and} \quad \boldsymbol{D}_{mn} = \boldsymbol{D}_{nm}. \tag{23}$$

Again, the skew-symmetry and semi-definiteness can be achieved by parameterization tricks:

$$\mathbf{\Lambda} = \frac{1}{2}(\tilde{\mathbf{\Lambda}} - \tilde{\mathbf{\Lambda}}^{\mathsf{T}}), \quad \text{and} \qquad \boldsymbol{D} = \tilde{\boldsymbol{D}}\tilde{\boldsymbol{D}}^{\mathsf{T}}, \tag{24}$$

where $\tilde{\mathbf{\Lambda}} \in \mathbb{R}^{n_{\boldsymbol{x}} \times n_{\boldsymbol{x}} \times D_1}$ and $\tilde{\boldsymbol{D}} \in \mathbb{R}^{D_1 \times D_2}$ contain learnable entries, and $(\tilde{\mathbf{\Lambda}}^{\mathsf{T}})^m_{\alpha\beta} = \tilde{\mathbf{\Lambda}}^m_{\beta\alpha}$. With this parameterization, one can easily verify that $[S, S] = \boldsymbol{\zeta}_{\alpha\beta,\mu\nu} \frac{\partial S}{\partial \boldsymbol{x}_\alpha} \frac{\partial E}{\partial \boldsymbol{x}_\beta} \frac{\partial S}{\partial \boldsymbol{x}_\mu} \frac{\partial E}{\partial \boldsymbol{x}_\nu} \geq 0$ and $[E, S] = \boldsymbol{\zeta}_{\alpha\beta,\mu\nu} \frac{\partial E}{\partial \boldsymbol{x}_\alpha} \frac{\partial E}{\partial \boldsymbol{x}_\beta} \frac{\partial S}{\partial \boldsymbol{x}_\mu} \frac{\partial E}{\partial \boldsymbol{x}_\nu} = 0$. Contracting the second and the fourth dimensions gives the $\boldsymbol{M}$ matrix, $\boldsymbol{M}(\boldsymbol{x}) = \boldsymbol{\zeta}_{\alpha\beta,\mu\nu} \frac{\partial E}{\partial \boldsymbol{x}_\beta} \frac{\partial E}{\partial \boldsymbol{x}_\nu}$.

### C.2. Modification for this Work

We apply three key assumptions following practices in the literature (Zhang et al., 2022; Lee et al., 2022) to make the problem more tractable. Similar to HNNs, we assume the GENERIC model forms under consideration are governed by the canonical Poisson matrix, that is, $\boldsymbol{L} = \begin{bmatrix} 0 & \boldsymbol{I}_N & 0 \\ -\boldsymbol{I}_N & 0 & 0 \\ 0 & 0 & 0 \end{bmatrix}$, where $N$ denotes the degrees of freedom in the system and $\boldsymbol{I}_N$ is the identify matrix of size $N$. This is equivalent to assuming that the reversible dynamics in the configuration space are foliated into canonical symplectic leaves $\{S = c\}$ for constant $c$. We also assume that the total entropy is expressed as a sum $S = \sum_i S_i$ in terms of some observable variables $S_i$. Lastly, we assume the energy function is decomposable similar to the reversible dynamics, that is, the network approximation takes the form $E(\boldsymbol{x}; \boldsymbol{\Theta}_E) = E_{qp}(\boldsymbol{q}, \boldsymbol{p}; \boldsymbol{\Theta}_{E_{qp}}) + E_S(\boldsymbol{S}; \boldsymbol{\Theta}_{E_S})$, where $E_{qp}$ and $E_S$ are parameterized as MLPs with parameters $\boldsymbol{\Theta}_{E_{qp}}$ and $\boldsymbol{\Theta}_{E_S}$), respectively.

Under these modeling assumptions, we modulate the kinetic and potential energy function $E_{qp}$ using the various modulation techniques described in Appendix A. Furthermore, we also modulate the learnable friction matrix through its symmetric core, i.e., $\boldsymbol{D} = \boldsymbol{D}_{\text{base}} + \boldsymbol{D}^{(k)}$, where the individual model parameters $\boldsymbol{D}^{(k)}$ are inferred from the hypernetwork.

## D. Proofs and Omitted Results

Here we record proofs and theoretical results omitted from the main body. First, we prove Proposition 3.1.

*[Proof of Proposition 3.1.* ] Let $\boldsymbol{F}_{\boldsymbol{x}}(\boldsymbol{\mu}) = \boldsymbol{f}(\boldsymbol{x}, \boldsymbol{\mu})$. For any $\mu \in P$, the Jacobian $\boldsymbol{F}'_{\boldsymbol{x}}(\boldsymbol{\mu})$ is a $n \times p$ matrix of rank at most $r \leq \min(n, p)$ (by assumption). If $\mu$ is a critical point, the dimension of the map $\boldsymbol{f}$ is 0 at $\boldsymbol{\mu}$. Otherwise, $\boldsymbol{F}'_{\boldsymbol{x}}(\boldsymbol{\mu})$ has constant rank $r' \leq r$ in a neighborhood of $\boldsymbol{\mu}$ by assumption, and the Constant Rank Theorem applies to show that $\boldsymbol{F}_{\boldsymbol{x}}$ is at most $r$ dimensional around $\boldsymbol{\mu}$. $\square$

Now, we show that the modulation applied in this work does not affect the structure-preservation guarantees of the employed Hamiltonian and GENERIC modeling strategies.

**Proposition D.1.** *Suppose $\mathcal{H}_{\Theta_H}, \boldsymbol{L}_{\Theta_L}, M_{\Theta_M}, E_{\Theta_E}, S_{\Theta_S}$ are as described. Then, modulation of the parameters $\Theta_i$ ($i \in \{H, L, M, E, S\}$) is compatible with energy conservation in the Hamiltonian case and the first two laws of thermodynamics in the GENERIC case.*

*Proof.* Modulation can be considered as a map $\Theta_i \mapsto \Theta'_i$ which accounts for the effect of unknown system parameters on the current functional data. For the Hamiltonian neural network, observe that

$$\frac{d\mathcal{H}_{\Theta'_H}}{dt} = \left(\frac{\partial \mathcal{H}_{\Theta'_H}}{\partial \boldsymbol{q}}\right)^{\mathsf{T}} \frac{d\boldsymbol{q}}{dt} + \left(\frac{\partial \mathcal{H}_{\Theta'_H}}{\partial \boldsymbol{p}}\right)^{\mathsf{T}} \frac{d\boldsymbol{p}}{dt} = \left(\frac{\partial \mathcal{H}_{\Theta'_H}}{\partial \boldsymbol{q}}\right)^{\mathsf{T}} \frac{\partial \mathcal{H}_{\Theta'_H}}{\partial \boldsymbol{p}} - \left(\frac{\partial \mathcal{H}_{\Theta'_H}}{\partial \boldsymbol{p}}\right)^{\mathsf{T}} \frac{\partial \mathcal{H}_{\Theta'_H}}{\partial \boldsymbol{q}} = 0,$$

establishing the conclusion in the Hamiltonian case.

For the metriplectic/GENERIC case, observe that

$$\boldsymbol{L}_{\Theta'_L} = \boldsymbol{\xi}_{\alpha\beta\gamma} \frac{\partial S_{\Theta'_S}}{\partial \boldsymbol{x}_\gamma}, \qquad \boldsymbol{M}_{\Theta'_M} = \boldsymbol{\zeta}_{\alpha\beta,\mu\nu} \frac{\partial E_{\Theta'_E}}{\partial \boldsymbol{x}_\beta} \frac{\partial E_{\Theta'_E}}{\partial \boldsymbol{x}_\nu}.$$

This implies the degeneracy conditions

$$[\boldsymbol{L}_{\Theta'_L}(\boldsymbol{x})\nabla S_{\Theta'_S}(\boldsymbol{x})]_\alpha = \boldsymbol{\xi}_{\alpha\beta\gamma}\frac{\partial S_{\Theta'_S}}{\partial \boldsymbol{x}_\gamma}\frac{\partial S_{\Theta'_S}}{\partial \boldsymbol{x}_\beta} = \boldsymbol{\xi}_{\alpha\gamma\beta}\frac{\partial S_{\Theta'_S}}{\partial \boldsymbol{x}_\beta}\frac{\partial S_{\Theta'_S}}{\partial \boldsymbol{x}_\gamma} = -\boldsymbol{\xi}_{\alpha\beta\gamma}\frac{\partial S_{\Theta'_S}}{\partial \boldsymbol{x}_\gamma}\frac{\partial S_{\Theta'_S}}{\partial \boldsymbol{x}_\beta} = 0,$$

$$[\boldsymbol{M}_{\Theta'_M}(\boldsymbol{x})\nabla E_{\Theta'_E}(\boldsymbol{x})]_\alpha = \boldsymbol{\zeta}_{\alpha\beta,\mu\nu}\frac{\partial E_{\Theta'_E}}{\partial \boldsymbol{x}_\beta}\frac{\partial E_{\Theta'_E}}{\partial \boldsymbol{x}_\nu}\frac{\partial E_{\Theta'_E}}{\partial \boldsymbol{x}_\mu} = -\boldsymbol{\zeta}_{\alpha\beta,\mu\nu}\frac{\partial E_{\Theta'_E}}{\partial \boldsymbol{x}_\beta}\frac{\partial E_{\Theta'_E}}{\partial \boldsymbol{x}_\nu}\frac{\partial E_{\Theta'_E}}{\partial \boldsymbol{x}_\mu} = 0,$$

where the last equality of the second line has swapped $\mu$ with $\nu$ and applied antisymmetry. Therefore the first two laws of thermodynamics in differential form hold as well. $\qquad\square$

# E. Datasets and Data Generation

In Table 5, we list the experimental setup used in generating data. For sampling trajectories of Hamiltonian/GENERIC systems, we randomly sample input parameters and initial conditions, solving initial value problems (IVPs) with the specified step size and terminal time. For solving these IVPs, we use the Runge–Kutta 4 (RK4) time integrator (Runge, 1895).

*Table 5.* Experimental setup: input parameter ranges, initial condition sampling ranges, step size, and terminal time.

| System | Input parameters | Initial conditions | Step size (sec) | Terminal time (sec) |
|---|---|---|---|---|
| **Mass spring** | $(m,k) \in [1,5]^2$ | $(q^0, p^0) \in [-9,-9]^2$ | 0.1 | 3 |
| **Pendulum** | $(m,k) \in [1,5]^2$ | $(q^0, p^0) \in [-2\pi, 2\pi] \times [-19,-19]$ | 0.1 | 3 |
| **Duffing** | $(\alpha, \beta) \in [2,5] \times [-5,-2]$ | $(q^0, p^0) \in [-3,3] \times [-2,2]$ | 0.1 | 3 |
| **Kepler** | $(m_1, m_2) \in [0.5, 2.5]^2$ | $(r, \theta) \in [2,3] \times [0, 2\pi]$ | 0.05 | 5 |
| **DNO** | $(m, \gamma) \in [1, 1.5] \times [0.05, 0.15]$ | $(q^0, p^0) \in [-2\pi, 2\pi] \times [-2, 2], S_0 = 0$ | 0.01 | 1 |

## E.1. Energy Conserving Systems

We now describe the considered energy conserving systems. As described in Section 2.1, the dynamics of the following systems are defined in terms of the symplectic gradient of their associated Hamiltonian function, i.e., as

$$\begin{bmatrix} \frac{d\boldsymbol{q}}{dt} \\ \frac{d\boldsymbol{p}}{dt} \end{bmatrix} = \begin{bmatrix} \frac{\partial \mathcal{H}}{\partial \boldsymbol{p}} \\ -\frac{\partial \mathcal{H}}{\partial \boldsymbol{q}} \end{bmatrix}.$$

**Mass Spring** The first system we consider is a mass-spring system, which is characterized by two physical parameters, the mass $m$ and the spring constant $k$, i.e., $\boldsymbol{\mu} = [m, k]$. The Hamiltonian of the system is given by

$$\mathcal{H}(q, p; m, k) = \frac{p^2}{2m} + \frac{kq^2}{2}. \tag{25}$$

**Pendulum** The next system is a pendulum system, which is characterized by two physical parameters, the mass $m$ and the pendulum length $l$, i.e., $\boldsymbol{\mu} = [m, l]$. The Hamiltonian of the system is given by

$$\mathcal{H}(q, p; m, l) = \frac{p^2}{2ml^2} + mgl(1 - \cos(q)). \tag{26}$$

This is derived from the Lagrangian

$$L(\theta, \dot{\theta}) = \frac{1}{2}ml^2\dot{\theta}^2 - mgl(1 - \cos\theta), \tag{27}$$

where the canonical momentum is given by $p = \frac{\partial L}{\partial \dot{\theta}} = ml^2\dot{\theta}$. The Hamiltonian is then obtained through the Legendre transform $H(q, p) = p\dot{\theta} - L$. Substituting $\dot{\theta} = p/(ml^2)$ yields the Hamiltonian in Eq. (26).

**Duffing oscillator** The third system is a duffing oscillator, which is characterized by two physical parameters, stiffness $\alpha$ and level of nonlinearity $\beta$, i.e., $\boldsymbol{\mu} = [\alpha, \beta]$. The Hamiltonian of the system is given by

$$\mathcal{H}(q, p; \alpha, \beta) = \frac{1}{2}p^2 + \frac{\alpha}{4}q^4 + \frac{\beta}{2}q^2. \tag{28}$$

**Kepler system.** The fourth system is the Kepler problem, defined in canonical coordinates $x = (q_1, q_2, p_1, p_2)$. The dynamics are given by

$$\dot{q}_1 = \frac{p_1}{m_2}, \qquad\qquad \dot{q}_2 = \frac{p_2}{m_2}, \qquad\qquad (29)$$

$$\dot{p}_1 = -\frac{m_1 m_2(q_1 - q_{0,1})}{r^3}, \qquad\qquad \dot{p}_2 = -\frac{m_1 m_2(q_2 - q_{0,2})}{r^3}, \qquad\qquad (30)$$

where

$$r = \sqrt{(q_1 - q_{0,1})^2 + (q_2 - q_{0,2})^2}. \qquad\qquad (31)$$

The corresponding Hamiltonian is

$$H(q, p) = \frac{p_1^2 + p_2^2}{2m_2} - \frac{m_1 m_2}{r}. \qquad\qquad (32)$$

### E.2. Dissipative Systems

Next, we describe the dissipative system considered in this study.

**Damped nonlinear oscillator** The damped nonlinear oscillator has the equations of motion

$$\frac{\mathrm{d}q}{\mathrm{d}t} = \frac{p}{m}, \qquad \frac{\mathrm{d}p}{\mathrm{d}t} = k\sin(q) - \gamma p, \qquad \frac{\mathrm{d}S}{\mathrm{d}t} = \frac{\gamma p^2}{mT}, \qquad\qquad (33)$$

where $(q, p)$ denotes the position and momentum of the particle, and $S$ is the entropy of the surrounding thermal bath. The constant parameters $m$, $\gamma$, and $T$ represent the mass of the particle, the damping rate, and the constant temperature of the thermal bath. In our study, we consider a parameterization in terms of $\mu = [m, \gamma]$.

The total energy of the damped nonlinear oscillator system is

$$E(q, p, S) = \mathcal{H}(q, p) + TS = \frac{p^2}{2m} - k\cos(q) + TS, \qquad\qquad (34)$$

where $\mathcal{H}(q, p)$ is the Hamiltonian of the oscillating particle (the sum of its kinetic and the potential energies). With this definition of the energy, the dynamics can be expressed in the GENERIC form, $\frac{\mathrm{d}\boldsymbol{x}}{\mathrm{d}t} = \boldsymbol{L}\frac{\partial E}{\partial \boldsymbol{x}} + \boldsymbol{M}\frac{\partial S}{\partial \boldsymbol{x}}$ with $\boldsymbol{x} = [q, p, S]^{\mathsf{T}}$, where the skew-symmetric and SPSD matrices, $\boldsymbol{L}$ and $\boldsymbol{M}$, are defined as

$$\boldsymbol{L} = \begin{bmatrix} 0 & 1 & 0 \\ -1 & 0 & 0 \\ 0 & 0 & 0 \end{bmatrix}, \qquad \text{and} \qquad \boldsymbol{M}(\boldsymbol{x}) = \begin{bmatrix} 0 & 0 & 0 \\ 0 & \gamma mT & -\gamma p \\ 0 & -\gamma p & \frac{\gamma p^2}{mT} \end{bmatrix}. \qquad\qquad (35)$$

## F. Baseline Methods

We consider the three representative optimization-based meta-learning algorithms: model-agnostic meta learning (MAML) (Finn et al., 2017), Reptile (Nichol et al., 2018), and almost no inner loop (ANIL) (Raghu et al., 2020). We can define a task, $\tau^{(k)}$, as a specific instance of system parameters, $\boldsymbol{\mu}^{(k)}$. All three methods consist of an *inner* loop and an *outer* loop. The inner loop takes $N_{\mathrm{in}}$ optimization gradient descent steps to update model parameters from the meta initial points $\theta_0^l$ given a training task $\tau^{(i)}$. Here, the result of this $N_{\mathrm{in}}$-step update is denoted as $\theta_{N_{\mathrm{in}}}(\theta_0^l, \tau^{(i)})$. Following this procedure, the outer loop updates the meta-learned initial points using the information obtained from the inner loop according to the one of the following rules:

$$\begin{aligned} \theta_0^{l+1} &= \theta_0^l - \beta \nabla_\theta L(\theta_{N_{\mathrm{in}}}(\theta, \tau^{(l)}))|_{\theta = \theta_0^l}, & \text{(MAML, ANIL)} \\ \theta_0^{l+1} &= \theta_0^l - \beta(\theta_{N_{\mathrm{in}}}(\theta_0^l, \tau^{(l)}) - \theta_0^l). & \text{(Reptile)} \end{aligned} \qquad (36)$$

Note that Reptile relies on only first-order gradient information, leading to efficient computation. ANIL operates the same as MAML, except that only the last layer is being updated in the inner loop while all the other model parameters are fixed. That is, during the inner loop, only the weight of the last layer, $W_L$, is updated.

# G. Implementation Details

All the code is developed using PYTHON and PYTORCH (Paszke et al., 2019). All experiments are conducted on a machine equipped with an Apple Silicon M4 chip (M4 Max, 128 GB memory). The base neural network is modeled as an MLP with 2 layers of 100 neurons each and SiLU activation. The hypernetwork is modeled as a single linear layer without nonlinearity.

The maximum number of training epochs is set to 10,000, and the number of auto-decoding gradient steps is set to 100. The learning rate for the outer-loop (i.e., for the meta-gradient update) is set to 0.001 and the one for the inner-loop is set to 0.002. The only exception are the rates for Reptile, which are set to 0.01 and 0.02, respectively, as the defaults result in diverging vector fields. The number of tasks in each inner loop (i.e., the number of dynamical systems with different parameters) is set to 5. We use the Adam optimizer (Kingma & Ba, 2014) for all gradient descent updates. For all methods and all configurations, we vary the random seed and repeat the same experiments five times.

To measure the field error, we consider the uniform mesh defined as follows: (Energy conserving systems) for Mass Spring, $(q, p) \in [-10, 10]^2$, for Pendulum $(q, p) \in [-6, 6] \times [-20, 20]$, for Duffing oscillator $(q, p) \in [-3, 3]^2$, with 100 equidistant points in each dimension. (Dissipative systems) for DNO, $(q, p, S) \in [-8, 8] \times [-1, 1] \times [0, 3]$ with 30 equidistant points in each dimension.

# H. Additional Experiments

### H.1. Performance Averaged over Varying Locality Constraints: $\lambda_\phi$ and $\lambda_z$

In this experiment, we examine the robustness of modulation-based meta-learning methods to the choice of locality regularization parameters. Specifically, we vary the regularization weights over $\lambda_\phi \in \{10^{-2}, 10^{-3}, 10^{-4}\}$ and $\lambda_z \in \{10^{-1}, 10^{-2}, 10^{-3}\}$, resulting in multiple combinations of locality strengths. For each $(\lambda_\phi, \lambda_z)$ pair, five independent runs are performed for all modulation methods. Model performance is first evaluated separately at each regularization setting (repeated 5 individual runs with different random seeds) and then averaged across all values of $\lambda_\phi$ and $\lambda_z$. This averaging procedure enables an assessment of both the mean performance and the variability induced by different locality constraints, rather than performance under a single tuned configuration.

The aggregated results are reported in Table 6. Overall, SVD-like modulation achieves consistently strong average performance while exhibiting relatively low variance across different regularization settings, indicating reduced sensitivity to the choice of locality parameters. In contrast, shift-based modulation shows noticeably larger variation, suggesting a higher dependence on careful tuning of $\lambda_\phi$ and $\lambda_z$. The remaining modulation methods demonstrate intermediate behavior, with moderate robustness but slightly weaker average performance compared to SVD-based approaches.

*Table 6.* [Energy-conserving systems] Model performance measured in the trajectory errors, $\epsilon_{\text{traj}}$ ($\downarrow$) and the field errors, $\epsilon_{\text{field}}$ ($\downarrow$) (lower values indicate better performance). The numbers indicate the mean ($\times 10^{-2}$) and the standard deviation (inside the parenthesis, $\times 10^{-3}$). For all values of $\lambda_\phi$ and $\lambda_z$, the results are obtained over 5 individual runs.

| Method | Duffing | | Mass Spring | | Pendulum | |
|--------|---------|---------|-------------|---------|----------|---------|
| | $\epsilon_{\text{field}}$ | $\epsilon_{\text{traj}}$ | $\epsilon_{\text{field}}$ | $\epsilon_{\text{traj}}$ | $\epsilon_{\text{field}}$ | $\epsilon_{\text{traj}}$ |
| FW | 10.24 (1.189) | 3.126 (2.039) | 2.211 (2.950) | 1.934 (2.988) | 9.35 (6.592) | 11.69 (10.66) |
| Shift | 9.450 (8.663) | 10.29 (31.79) | 23.52 (45.80) | 21.53 (51.02) | 11.73 (15.64) | 14.73 (16.64) |
| MR | 10.10 (1.276) | 2.812 (2.569) | 2.270 (5.803) | 2.020 (4.264) | 7.134 (6.472) | 8.785 (6.041) |
| RO | 10.23 (1.167) | 2.710 (2.311) | 2.157 (3.034) | 1.707 (3.090) | 8.018 (8.327) | 9.645 (10.44) |
| SVD | 10.03 (2.194) | 2.983 (8.726) | 2.101 (3.320) | 1.739 (4.107) | 6.318 (10.22) | 7.974 (17.14) |
| SVD(5) | 9.968 (1.956) | 2.995 (9.604) | 2.081 (4.617) | 1.670 (5.423) | 5.305 (4.204) | 6.554 (6.536) |

### H.2. Orthogonality Constraints

For SVD-like modulation, we additionally consider a soft orthogonality constraint on the learned basis vectors $U$ and $V$ in the low-rank decomposition. Specifically, we encourage the columns of $U$ and $V$ to remain approximately orthogonal by adding a regularization term weighted by $\lambda_{\text{ortho}}$ to the training objective. This constraint is intended to better align the learned decomposition with the interpretation of SVD-like bases and to reduce redundancy among modulation directions.

Figure 9 compares the performance of SVD(5) with and without the orthogonality constraint (i.e., $\lambda_{\text{ortho}} = 0.1$ and $\lambda_{\text{ortho}} = 0$, respectively). Circles ($\bullet$) denote results obtained with the orthogonality regularization enabled, while crosses ($\times$) correspond

to the unconstrained variant. Across all evaluated settings, the orthogonality-constrained version consistently achieves lower errors, indicating improved stability and effectiveness of the learned modulation. These results suggest that softly enforcing orthogonality provides additional benefits for SVD-like modulation without introducing restrictive hard constraints.

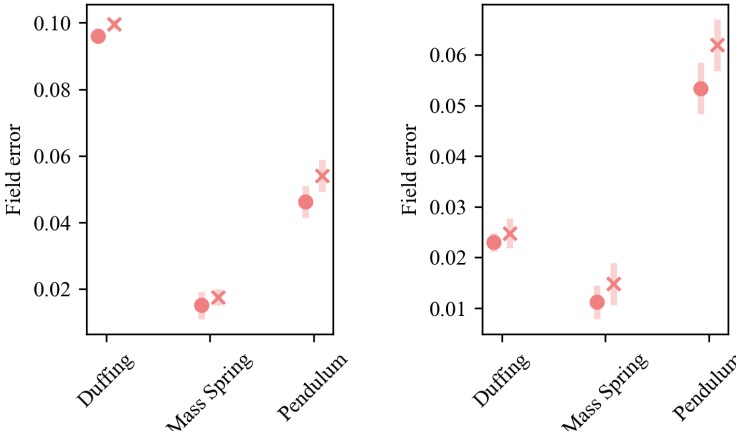

*Figure 9.* [Pendulum] The effect of the orthogonality constraints. The markers ● and × indicate with and without soft-constraining, respectively.

## H.3. Initialization Schemes for the Latent Codes

Table 7 reports the model performance measured in the field errors $\epsilon_{\text{field}}$ and trajectory errors $\epsilon_{\text{traj}}$; the models are trained with two different training algorithms: (1) the proposed algorithm, which takes the average of the latent vectors of the training instances to initialize the validation/test latent vectors, and (2) the traditional meta-learning algorithm, which initializes the validation/test latent vectors as zero vectors (i.e., a vector consisting of all zero elements). In Table 7, the results of the proposed algorithm are reported under 'Avg' columns and the those of the traditional algorithm are reported under 'Zero' columns. The proposed training algorithms consistently show the improved performance over the traditional zero-initialization scheme.

*Table 7.* [Energy-conserving systems] Model performance measured in the trajectory errors, $\epsilon_{\text{traj}}$ ($\downarrow$) and the field errors, $\epsilon_{\text{field}}$ ($\downarrow$) (lower values indicate better performance). The numbers indicate the mean ($\times 10^{-2}$) and the standard deviation (inside the parenthesis, $\times 10^{-3}$), obtained over 5 individual runs.

| | Init. | Duffing | | Mass Spring | | Pendulum | |
|---|---|---|---|---|---|---|---|
| | | $\epsilon_{\text{field}}$ | $\epsilon_{\text{traj}}$ | $\epsilon_{\text{field}}$ | $\epsilon_{\text{traj}}$ | $\epsilon_{\text{field}}$ | $\epsilon_{\text{traj}}$ |
| FW | Zero | 14.07 (10.4) | 12.20 (21.4) | 6.986 (2.96) | 7.441 (7.01) | 18.53 (19.7) | 23.08 (40.9) |
| | Ours | 10.30 (2.06) | 2.784 (1.70) | 1.600 (1.62) | 1.307 (2.86) | 8.231 (12.4) | 10.65 (2.70) |
| Shift | Zero | 11.21 (112.) | 15.08 (147.) | 32.87 (11.6) | 20.74 (8.61) | 27.12 (42.5) | 33.43 (40.9) |
| | Ours | 9.031 (2.15) | 5.952 (5.48) | 16.66 (16.6) | 13.18 (17.7) | 9.758 (9.77) | 12.88 (9.81) |
| MR | Zero | 11.40 (13.4) | 11.63 (3.63) | 9.661 (17.6) | 10.83 (4.19) | 20.47 (58.0) | 23.67 (74.5) |
| | Ours | 10.15 (0.69) | 2.395 (3.36) | 1.853 (9.83) | 1.387 (2.25) | 7.069 (7.37) | 8.323 (9.59) |
| RO | Zero | 12.12 (9.19) | 12.27 (5.52) | 10.70 (12.6) | 13.83 (31.5) | 16.60 (17.2) | 21.05 (22.4) |
| | Ours | 10.10 (0.25) | 2.593 (2.32) | 1.517 (1.21) | 1.370 (2.10) | 6.473 (7.52) | 8.271 (14.1) |
| SVD | Zero | 10.60 (5.04) | 11.22 (4.96) | 8.068 (14.5) | 9.547 (24.9) | 21.89 (69.7) | 26.47 (64.6) |
| | Ours | 10.18 (1.80) | 2.328 (1.80) | 1.511 (2.72) | 1.196 (3.26) | 5.088 (6.62) | 6.726 (12.4) |
| SVD(5) | Zero | 16.00 (16.9) | 14.28 (14.5) | 7.150 (12.3) | 7.975 (19.5) | 21.39 (69.4) | 24.46 (74.6) |
| | Ours | 10.03 (0.55) | 2.303 (1.84) | 1.509 (4.14) | 1.116 (3.34) | 4.624 (4.80) | 5.336 (5.06) |

## H.4. Experiments with Varying Ranks $r$ in SVD($r$)

In this experiment, we study the effect of the rank parameter $r$ in SVD-like modulation by varying $r \in \{1, 2, 3, 5, 7\}$. All experiments are conducted on the pendulum system while keeping the base network architecture, latent dimension, and training protocol fixed. By varying only the rank of the low-rank decomposition, this experiment isolates how the expressive capacity of SVD-like modulation depends on $r$.

Figure 10 summarizes the resulting performance across different ranks. Increasing the rank improves performance initially, indicating that additional modulation directions help capture task-specific variations. However, performance gains saturate beyond a moderate rank, with $r = 5$ consistently achieving the best or near-best results across metrics. Further increasing the rank to $r = 7$ does not lead to meaningful improvement and in some cases slightly degrades performance, suggesting diminishing returns from additional rank. These results indicate that a relatively small rank is sufficient for effective modulation, and that SVD(5) provides a favorable balance between expressivity, stability, and parameter efficiency.

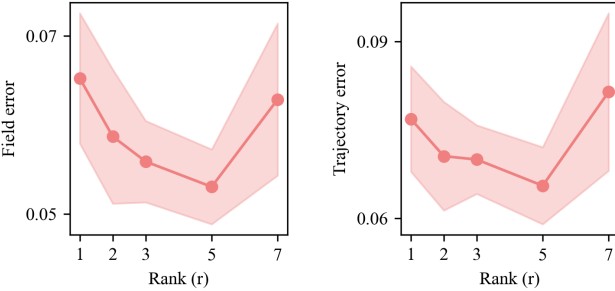

*Figure 10.* [Pendulum] Field and trajectory errors for SVD-like modulation with varying rank $r$.

## H.5. Sparsity Constraint on the Inferred Singular Values $\{d_i\}$

In the practical implementation of the SVD($r$) modulation, we introduce an additional sparsity-inducing constraint on the inferred singular values $\{d_i\}_{i=1}^r$. While the rank parameter $r$ specifies the maximum number of available modulation directions, not all directions are necessarily required to adapt to a given target system. To encourage an effective rank smaller than $r$, we impose an $\ell_1$ penalty on the vectorized singular values produced by the hyper-network. Specifically, the training objective is augmented with the regularization term $\lambda_{\mathrm{sp}} \sum_{i=1}^r |d_i|$, which promotes sparsity in $\{d_i\}$ and allows the model to automatically deactivate unnecessary modulation components. In all main-body experiments, we set $\lambda_{\mathrm{sp}} = 10^{-3}$. We further study the effect of varying $\lambda_{\mathrm{sp}}$ and analyze its influence on the learned effective rank and downstream performance.

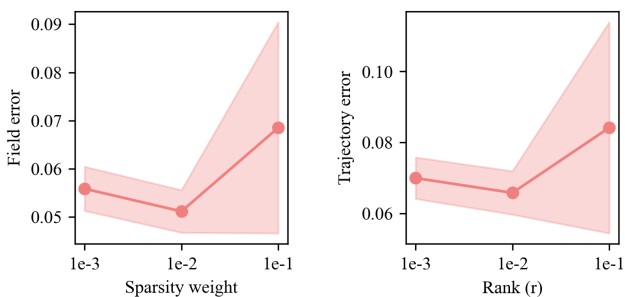

*Figure 11.* [Pendulum] Field and trajectory errors for SVD-like modulation with varying rank $\lambda_{\mathrm{sp}}$.

Using the pendulum system as a representative example, we evaluate the effect of the sparsity weight by considering $\lambda_{\mathrm{sp}} \in \{10^{-3}, 10^{-2}, 10^{-1}\}$. After test-time auto-decoding, we count the number of nonzero entries in the inferred singular value vector $\{d_i\}_{i=1}^r$ for each target system and report the average across all test instances. This results in effective ranks of 3.8, 3.12, and 2.48 for $\lambda_{\mathrm{sp}} = 10^{-3}, 10^{-2}$, and $10^{-1}$, respectively. Figure 11 shows the model performance for varying sparsity weights. Increased sparsity (1e-2) provide some additional benefits; that is, it encourages a compact set of active singular values helps remove redundant modulation directions. However, overly strong sparsity (1e-1) degrades performance,

as aggressively suppressing singular values reduces the expressive capacity of the modulation.

**Analysis on learned basis vectors $U$ and $V$**   Using SVD(5) with $\lambda_{\text{sp}} = 10^{-2}$ as a representative setting, we further examine the learned basis vectors $U$ and $V$ and their relationship to the underlying weight matrix. Specifically, we consider the weight matrix of an internal hidden layer, $W \in \mathbb{R}^{100 \times 100}$, where 100 denotes the layer width, together with the learned basis matrices $U = [u_1, \ldots, u_5]$ and $V = [v_1, \ldots, v_5]$, with $u_i, v_i \in \mathbb{R}^{100}$.

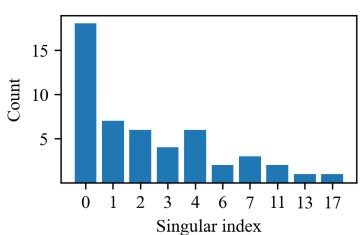

We compute the singular value decomposition of $W$ as $W = U_w \Sigma_w V_w^\top$ and identify, for each learned basis vector $u_i$ and $v_i$, the closest corresponding singular vectors in $U_w$ and $V_w$, respectively. This matching is performed by selecting the indices that maximize the absolute inner product, i.e., $\arg\max_j |u_i^\top (U_w)_j|$ and $\arg\max_j |v_i^\top (V_w)_j|$, $i = 1, \ldots, 5$.

Figure 12 summarizes the distribution of the matched singular vector indices aggregated over five independent runs and across both $U$ and $V$. The results show that the learned modulation bases most frequently align with the leading singular modes of $W$, while higher-index singular vectors (i.e., 11, 13, and 17) are selected only sporadically. This concentration on low-index modes indicates that, under the sparsity constraint, the SVD-based modulation effectively exploits a small subset of dominant directions in the base network, yielding an adaptive representation with reduced effective rank.

*Figure 12.* Indices of matching singular vectors

**Analysis of inferred singular values $\{d_i\}$**   The analysis above characterizes the set of candidate modulation directions learned through the basis matrices $U$ and $V$, which align with a small subset of the singular vectors of the base weight matrix $W$. We now examine the role of the inferred singular values $\{d_i\}_{i=1}^r$, which operate on top of this learned basis and determine which of these candidate directions are actively amplified for a given target system.

For each test instance, auto-decoding yields a sparse vector $\{d_i\}$ whose nonzero entries select a subset of the learned basis directions. On average, approximately $3.12$ entries are nonzero, indicating that adaptation typically amplifies only three to four basis directions. These selections are highly structured: the activated indices consistently correspond to a subset of the singular modes identified in Figure 12. In particular, most test instances primarily leverage basis vectors aligned with the leading singular directions of $W$, while directions associated with higher-index singular vectors are rarely activated.

### H.6. Gradient-subspace Analysis

The proposed low-rank modulation strategies are motivated by the observation that task-dependent variations in dynamical systems often lie in a low-dimensional subspace. A prior work (Kirchmeyer et al., 2022) has showed that adaptation directions in parameterized dynamical systems tend to concentrate in a low-dimensional gradient subspace. Defining the gradient subspace at the shared parameters $\Theta_{\text{base}}$ as

$$\mathcal{G}_{\Theta_{\text{base}}} = \text{Span}\Big(\big\{\nabla_\Theta L(\Theta_{\text{base}}, D^\mu)\big\}_{\mu \in \mathcal{P}}\Big), \tag{37}$$

it is shown that the adaptation directions of linearly parameterized dynamical systems with $n_\mu$ varying physical parameters lie in a low-dimensional subspace, i.e.,

$$\dim(\mathcal{G}_{\theta_c}) \leq n_\mu \ll \dim(\Theta). \tag{38}$$

While this guarantee is derived under linear parameterization, the same work further reports that the low-rank structure persists empirically for nonlinear dynamics (Appendix D), consistent with prior observations that gradient updates concentrate in a low-dimensional subspace during training.

To provide additional evidence in our setting, we perform the same gradient-subspace analysis by constructing

$$G = [g_1, \ldots, g_{N_{\text{test}}}], \qquad g_i = \text{vec}(\nabla_\Theta L(\Theta_{\text{base}}, D^{\mu_i})), \tag{39}$$

where each column corresponds to the gradient vector obtained from a different environment. We then examine the spectrum of the Gram matrix $G^\top G$. Across all benchmark systems, we observe strong spectral concentration: the top 5 eigenvalues account for approximately 99% of the explained variance. This indicates that most cross-instance variation is captured by a small number of dominant adaptation directions.

We further extend this analysis in a layer-wise manner. Rather than considering gradients with respect to all model parameters jointly, we restrict the gradients to the parameter subset of each individual layer and construct the corresponding layer-wise gradient matrix $G_\ell$. We then analyze the spectrum of $G_\ell^\top G_\ell$ for each layer. Across all considered systems (mass-spring, pendulum, and Duffing), we again observe consistent spectral concentration at the layer level, with the top five eigenvalues accounting for approximately 96.8% of the explained variance. This suggests that environment-induced variation remains low-dimensional even within individual layers.

Taken together, these results provide empirical support for the use of low-rank modulation. They suggest that both the full parameter space and the per-layer parameter subspaces can be well approximated by a small number of dominant directions, which is consistent with our SVD-like and multi-rank parameterizations.

### H.7. Kepler System

Table 8 reports trajectory errors on the Kepler problem. Due to the higher dimensionality of the system (4D phase space), we report only trajectory errors, as reliable field error estimation would require substantially denser sampling of the state space. Consistent with the previous experiments, modulation-based approaches significantly outperform optimization-based meta-learning baselines. Among all methods, the proposed SVD-like modulation achieves the best overall performance, with SVD obtaining the lowest trajectory error.

Results are averaged over locality regularization coefficients $\{10^{-1}, 10^{-2}, 10^{-3}\}$ and weight decay coefficients $\{10^{-2}, 10^{-3}, 10^{-4}\}$, with 5 runs per configuration.

We emphasize that further evaluation on more realistic settings, including higher-dimensional systems, PDEs, and noisy or partially observed observations, remains an important direction for future work.

*Table 8.* Trajectory errors on the Kepler system.

| Method | $\epsilon_{\text{traj}}$ |
|---|---|
| Scratch | 0.4573 |
| MAML | 0.2034 |
| ANIL | 0.3551 |
| Reptile | 0.4558 |
| FW | 0.0770 |
| Shift | 0.1775 |
| MR(5) | 0.0814 |
| RO | 0.0635 |
| SVD | **0.0516** |
| SVD(5) | 0.0548 |

### H.8. Phase Field Presentation

H.8.1. DUFFING

Figure 13 displays the phase space vector field representations of the ground-truth target field, and the reconstructed vector field using the learned Hamiltonian functions. For this benchmark problem, MAML and Reptile struggle to learn an accurate Hamiltonian and so produce a similar result to the model trained from scratch. ANIL produces a slightly improved reconstruction of the vector field, but still fails to capture fine details. All modulation-based methods outperform the optimization-based methods, in particular capturing details of the vector field in the range $q \in [-3, 3]$.

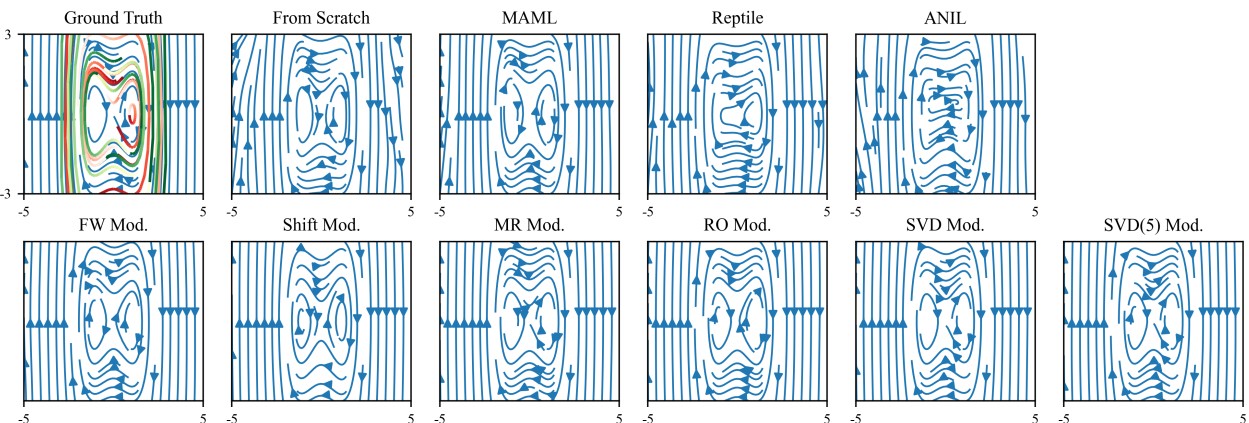

*Figure 13.* [Duffing oscillator] Illustration of phase space vector fields that are obtained through 300-shot adaptation to target fields during the test phase. All fields are depicted on $(q, p)$ space.

### H.8.2. MASS SPRING

Figure 14 depicts the phase space vector fields with $n_{\mathcal{T}} = 10$. The optimization-based methods demonstrate improved performance compared to learning the vector field from scratch. This demonstrates the power of meta-learning; given only the trajectories depicted in the red color, training from scratch struggles to capture the unseen areas of the vector field (even with structure-preserving modeling), while the meta-learned vector field allows accurate reconstructions of the unseen area. All modulation-based techniques also result in reconstructions that are aligned well with the target field.

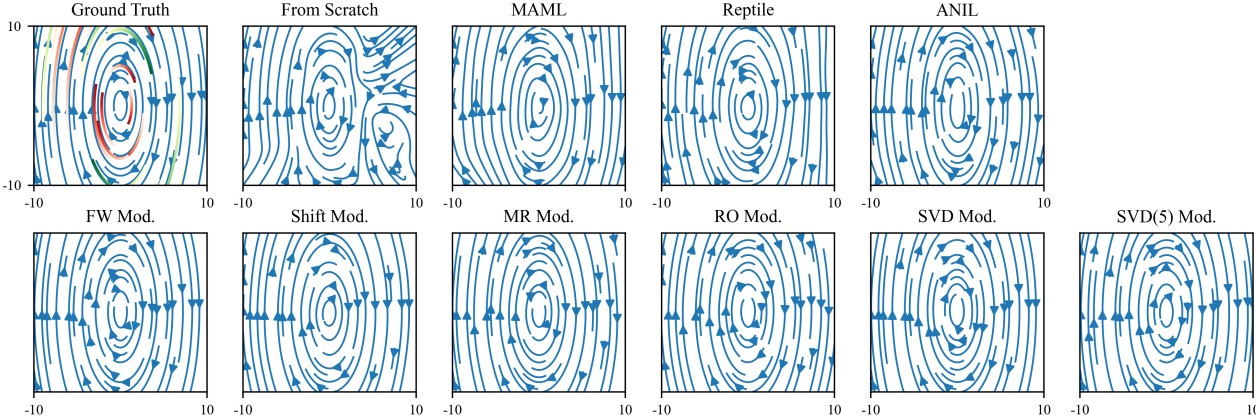

*Figure 14.* [Mass Spring] Illustration of phase space vector fields that are obtained through 300-shot adaptation to target fields during the test phase. All fields are depicted on the $(q, p)$ space.

### H.8.3. PENDULUM

Figure 15 depicts the phase space vector fields with $n_{\mathcal{T}} = 10$. Similar to the results of the previous benchmark problems, the optimization-based methods demonstrate improved performance compared to learning the vector field from scratch. Also, all modulation-based techniques also result in reconstructions that are aligned well with the target field.

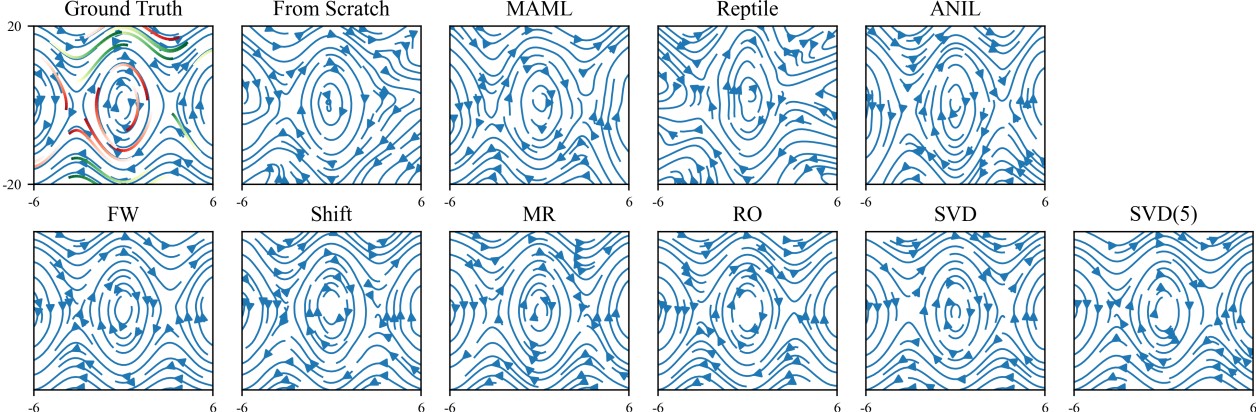

*Figure 15.* [Pendulum] Illustration of phase space vector fields that are obtained through 300-shot adaptation to target fields during the test phase. All fields are depicted on the $(q, p)$ space.

## H.9. Latent Space Visualization

To investigate whether the learned latent representations capture meaningful physical structure, we visualize the latent space learned for the pendulum system and examine its relationship to the underlying physical parameters, namely the mass $m$ and length $l$. We consider the SVD(5) model on the pendulum system. We uniformly sample 100 points from the $(l, m) \in [1, 5]^2$

parameter space and, for each sample, perform auto-decoding using the trained base model while updating only the latent variables. The latent dimension is set to two (i.e., dim($z$)=2) as the intrinsic dimension for the physical parameter space for the pendulm problem is 2 (the length and the mass) and also to facilitate direct visualization.

Figure 16 shows the resulting latent representations visualized using a triangulated surface plot. The learned latent space exhibits a smooth and structured organization that correlates with variations in the physical parameters. In particular, increasing values of $l$ and $m$ are associated with systematic changes in the latent coordinates, with one latent dimension showing an increasing trend while the other varies in an opposing manner. This suggests that the learned latent space encodes physically meaningful information and reflects the underlying parameter space in a coherent and interpretable way, despite not being explicitly constrained to do so during training.

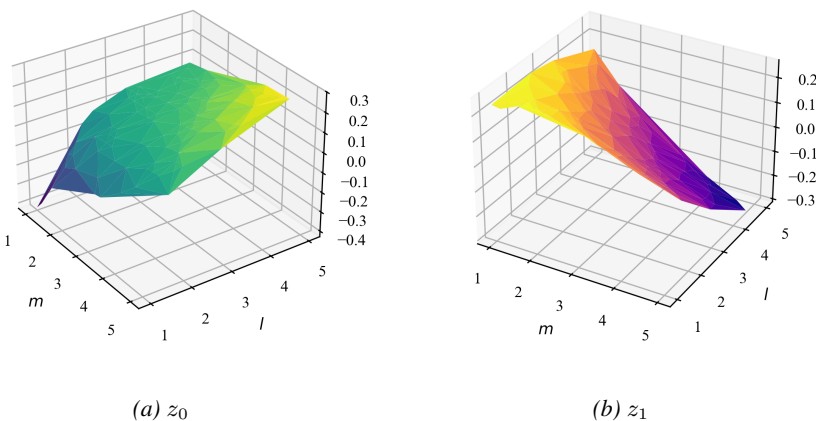

(a) $z_0$           (b) $z_1$

*Figure 16.* [Pendulum] Latent space visualization

**Interpolation in the learned latent space**    To further assess the structure and smoothness of the learned latent space, we perform an interpolation experiment using the same 100 sampled points from the $(m, l)$ parameter space of the pendulum system (Figure 17). We randomly split these samples into 50 points (●) used to construct an interpolant in the latent space and 50 held-out points (●) used for evaluation. The interpolant is built using the latent codes obtained via auto-decoding, and predictions at the held-out parameter values are generated by interpolating in the latent space rather than performing additional auto-decoding.

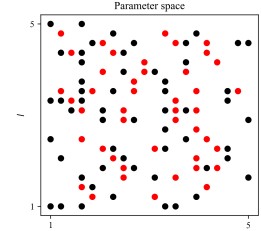

*Figure 17.* $(m, l)$-parameter space.

Figure 18 reports the resulting interpolation performance. The interpolated latent representations produce accurate predictions at unseen parameter locations, indicating that the learned latent space varies smoothly with respect to the underlying physical parameters. Moreover, the interpolation results preserve the qualitative structure observed in the direct latent visualization, suggesting that the latent space reflects the geometry of the physical parameter space. These findings provide further evidence that the modulation-based latent representation captures meaningful and smoothly varying physical information, enabling effective interpolation across parameter regimes.

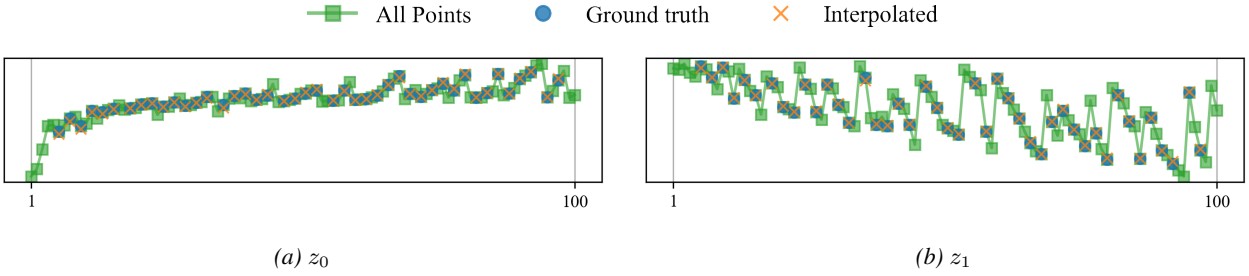

(a) $z_0$           (b) $z_1$

*Figure 18.* [Pendulum] Interpolation in the learned latent space

## H.10. Performance for Varying Latent Dimensions

We study the effect of the latent dimensionality by varying the size of the latent vector $z$. Figure 19 reports the results; as the latent dimension increases, performance improves initially, indicating that additional latent capacity enables more expressive modulation. However, beyond moderate sizes (approximately $z = 5$), further increases in latent dimension do not lead to meaningful or consistent performance gains, suggesting saturation of latent expressivity. Across all latent dimensions considered, SVD-based modulation methods, particularly SVD and SVD(5), consistently achieve the best performance. Notably, SVD(5) attains near-optimal performance with a small latent dimension, highlighting its parameter efficiency and effective use of latent capacity.

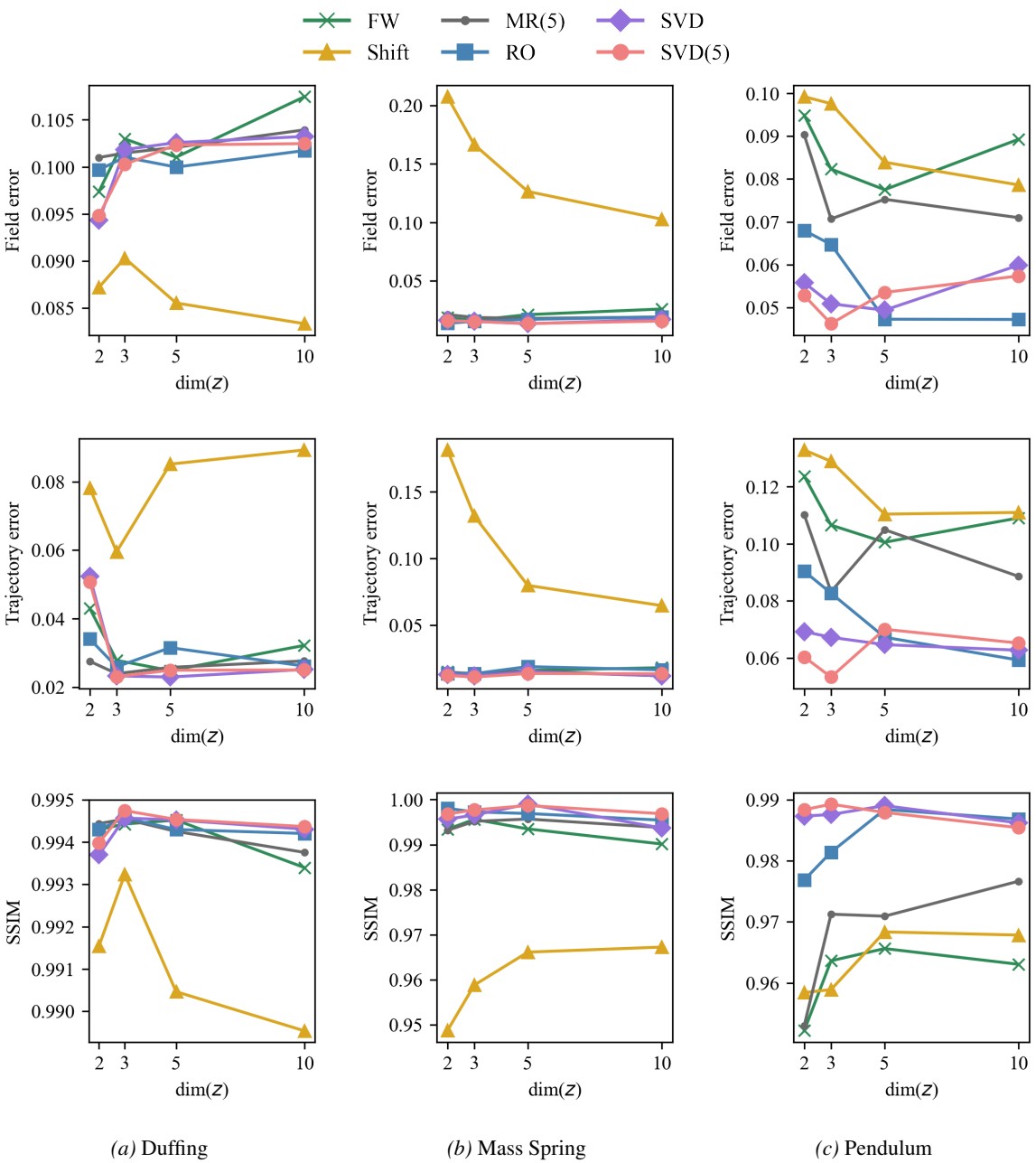

*(a)* Duffing       *(b)* Mass Spring      *(c)* Pendulum

*Figure 19.* [Energy conserving systems] Field errors (↓), trajectory errors (↓), and SSIM (↑) measured for all systems with varying latent dimensions (dim($z$)).

## H.11. Adaptation to the Target Field During Test Phase via Auto-decoding

Figures 20–22 illustrate 0-shot field (i.e., the meta-learned field with no update steps), $\{5,10,25,50\}$-shot updated vector field, and a target field, chosen from the test set for the mass spring, pendulum, and Duffing oscillators problems. The red gradient lines depict trajectories from the test set $\mathcal{T}_{\text{test}}^{(k)}$ that are used as target data in the auto-decoding process described in Algorithm 2.

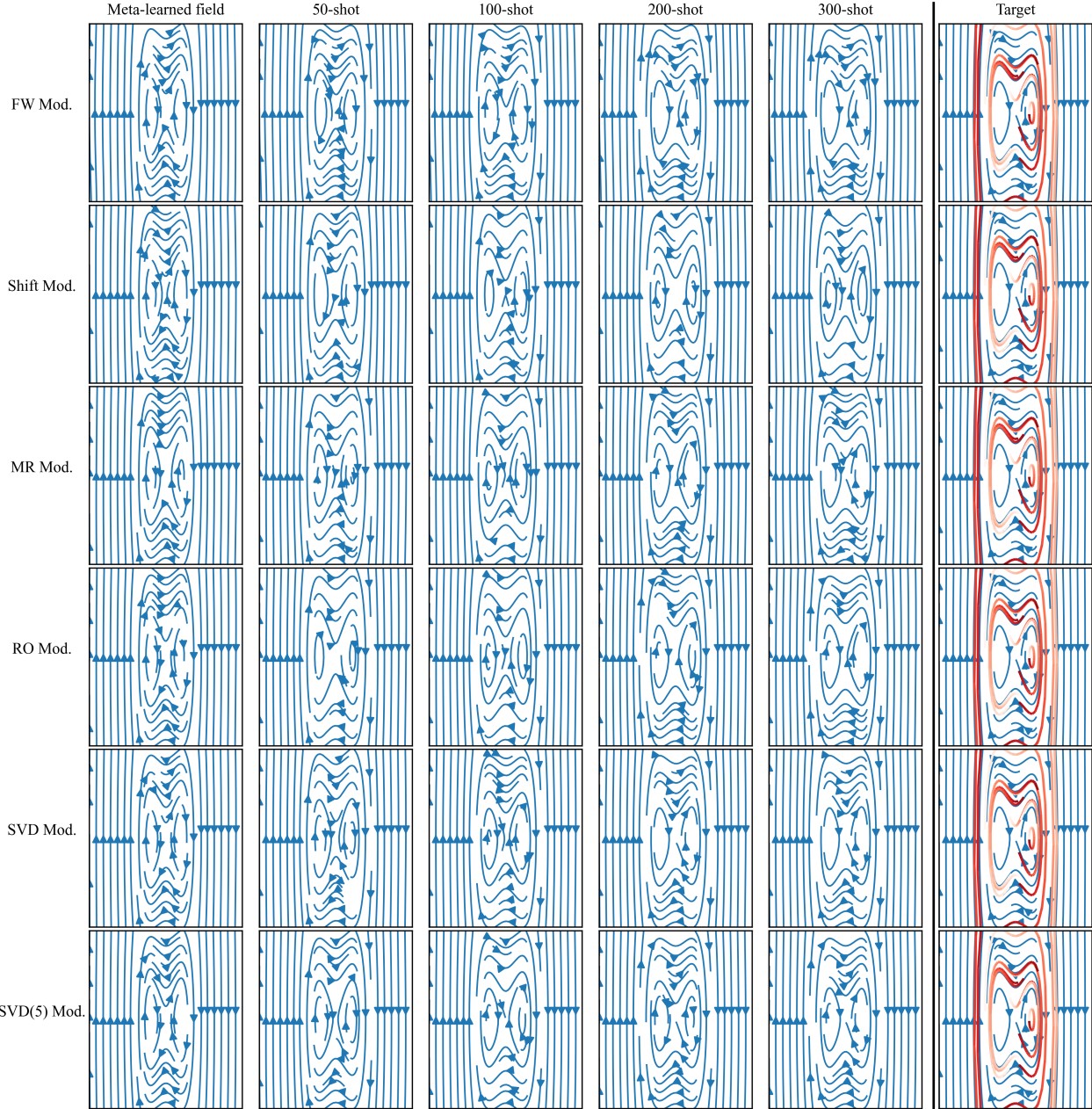

*Figure 20.* [Duffing] Illustration of phase space vector fields that are obtained through $\{0,50,100,200,300\}$-shot adaptation to target fields during the test phase.

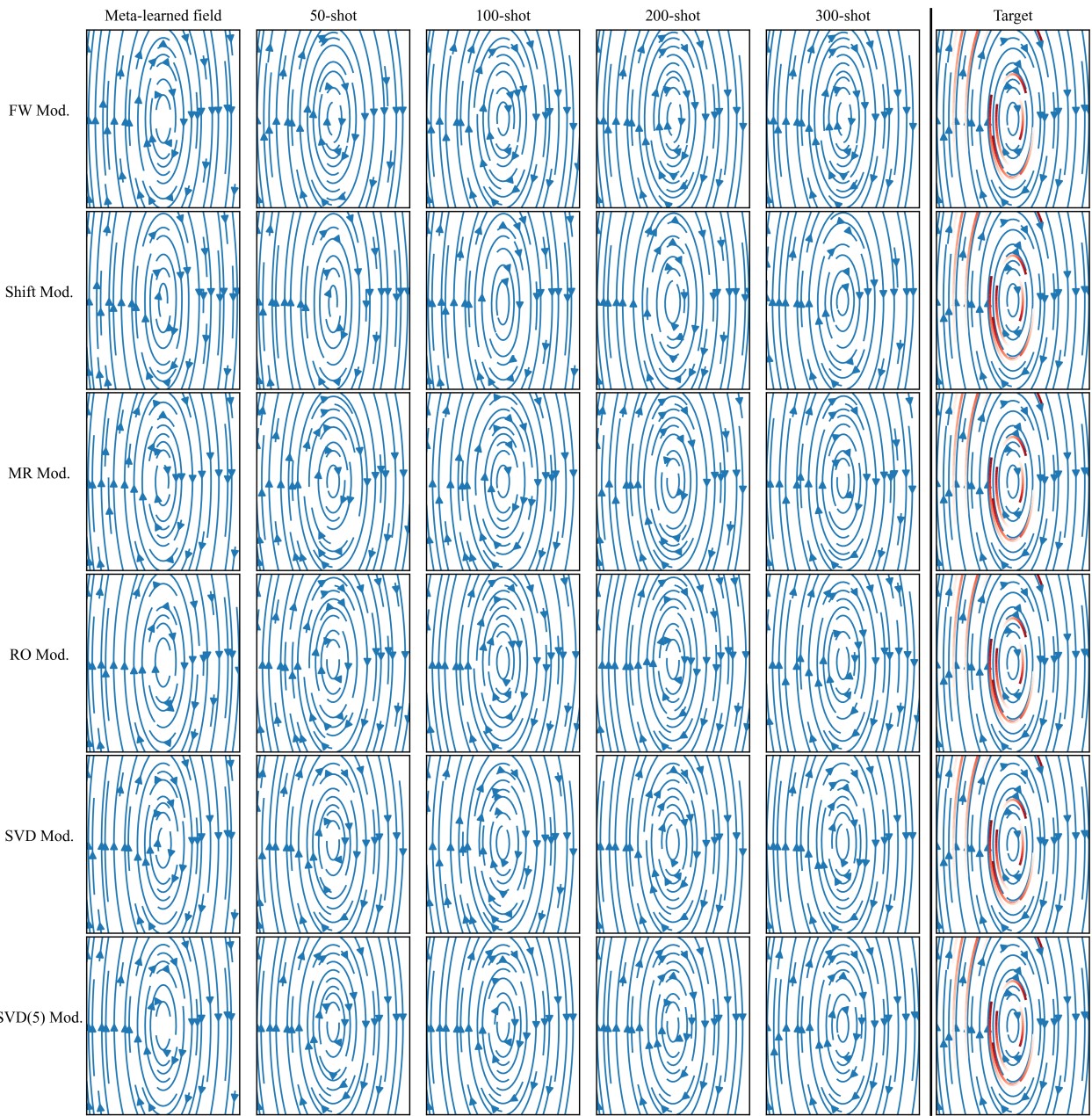

*Figure 21.* [Mass Spring] Illustration of phase space vector fields that are obtained through {0,50,100,200,300}-shot adaptation to target fields during the test phase.

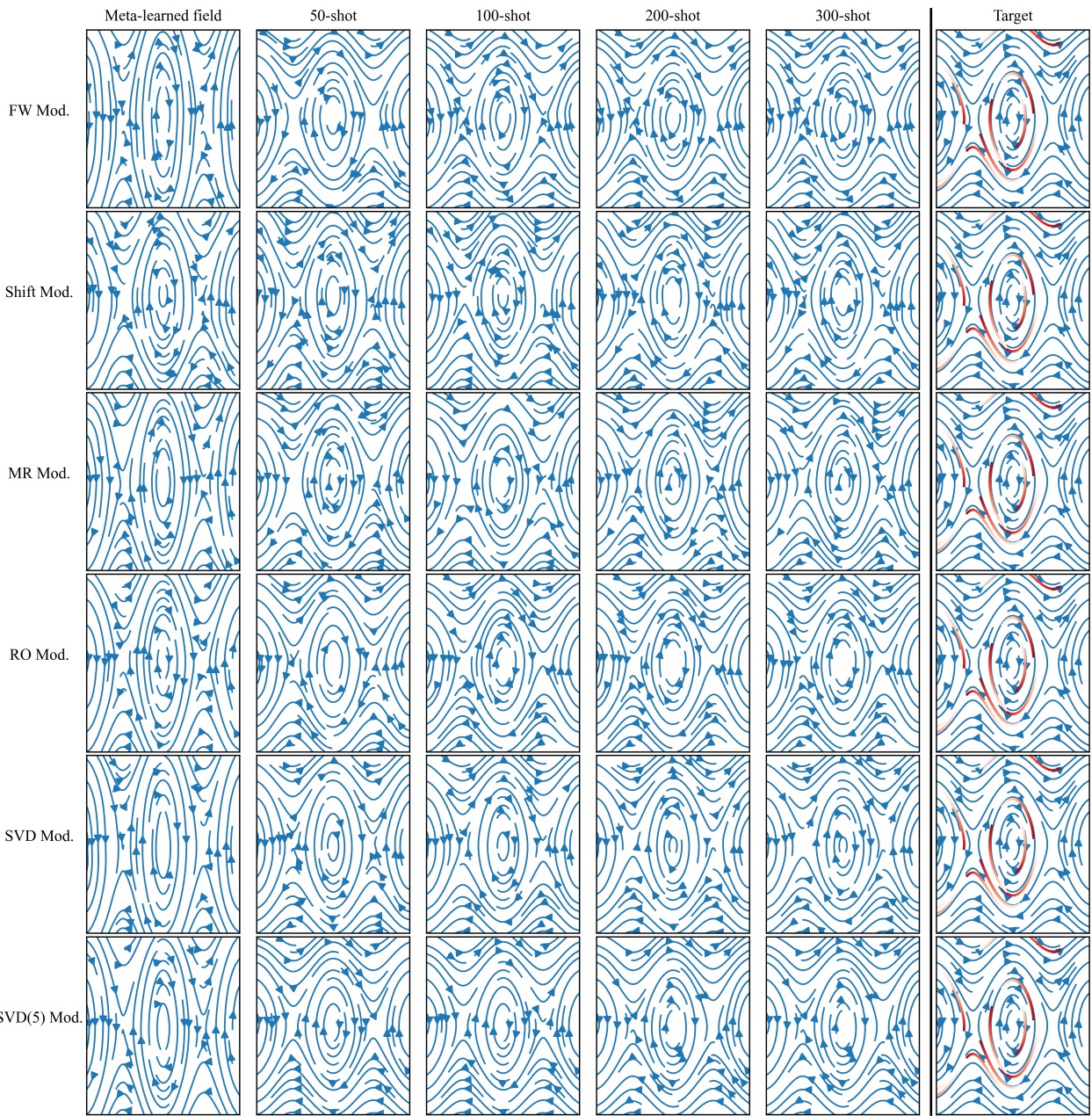

*Figure 22.* [Pendulum] Illustration of phase space vector fields that are obtained through {0,50,100,200,300}-shot adaptation to target fields during the test phase.

## H.12. Multi-domain Evaluation Experiments

In this experiment, we evaluate the ability of modulation-based methods to generalize across multiple dynamical systems using a shared base network. Unlike previous experiments where a separate base model is trained for each system, here a single base network is trained using a combined dataset consisting of trajectories from the pendulum, mass-spring, and Duffing systems. System-specific adaptation during testing is performed exclusively through latent modulation, without modifying the shared base parameters. This setting provides a stringent test of the expressivity and generalization capability of the modulation mechanisms.

The results are summarized in Figure 23. Overall, modulation-based approaches are able to adapt a shared model across heterogeneous systems, demonstrating the flexibility of latent modulation. Among all methods, SVD-based modulation, including SVD/SVD(5), consistently achieves the strongest performance across both field and trajectory error metrics. These results indicate that the structured low-rank decomposition employed by SVD-like modulation provides sufficient expressivity to capture system-specific characteristics while maintaining robustness in a multi-domain setting.

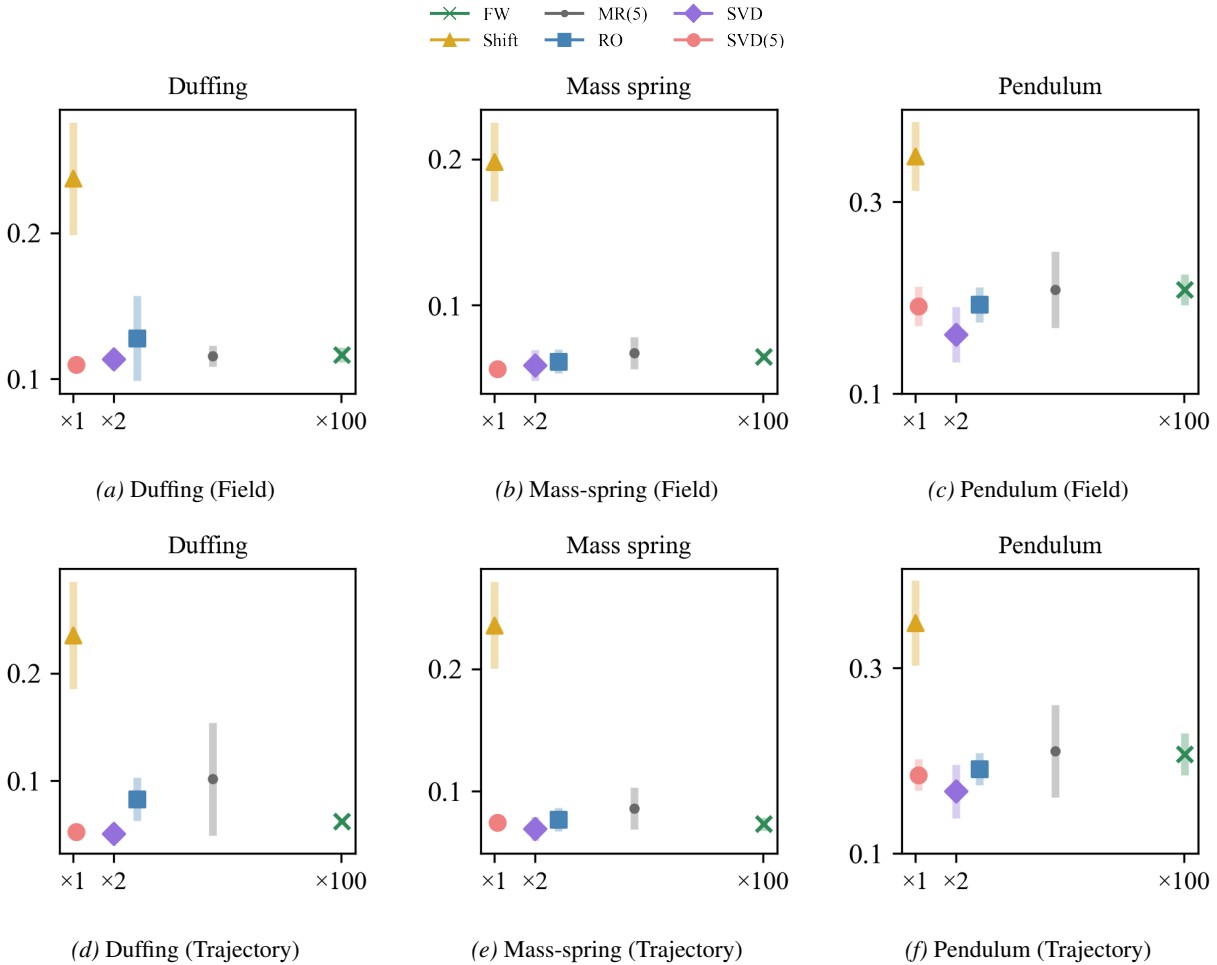

*Figure 23.* [Multi-domain] Field and trajectory errors versus relative hyper-network size.

## H.13. Dissipative System Visualization

In Figure 24, we present results that are obtained with DNO. The top panels show the target velocity field with $S = 1$ and the learned vector fields by using MAML and all modulation methods; the results of Reptile and ANIL are omitted as they do not produce meaningful results. The bottom panels show the target energy function and the learned energy functions by using the modulation methods.

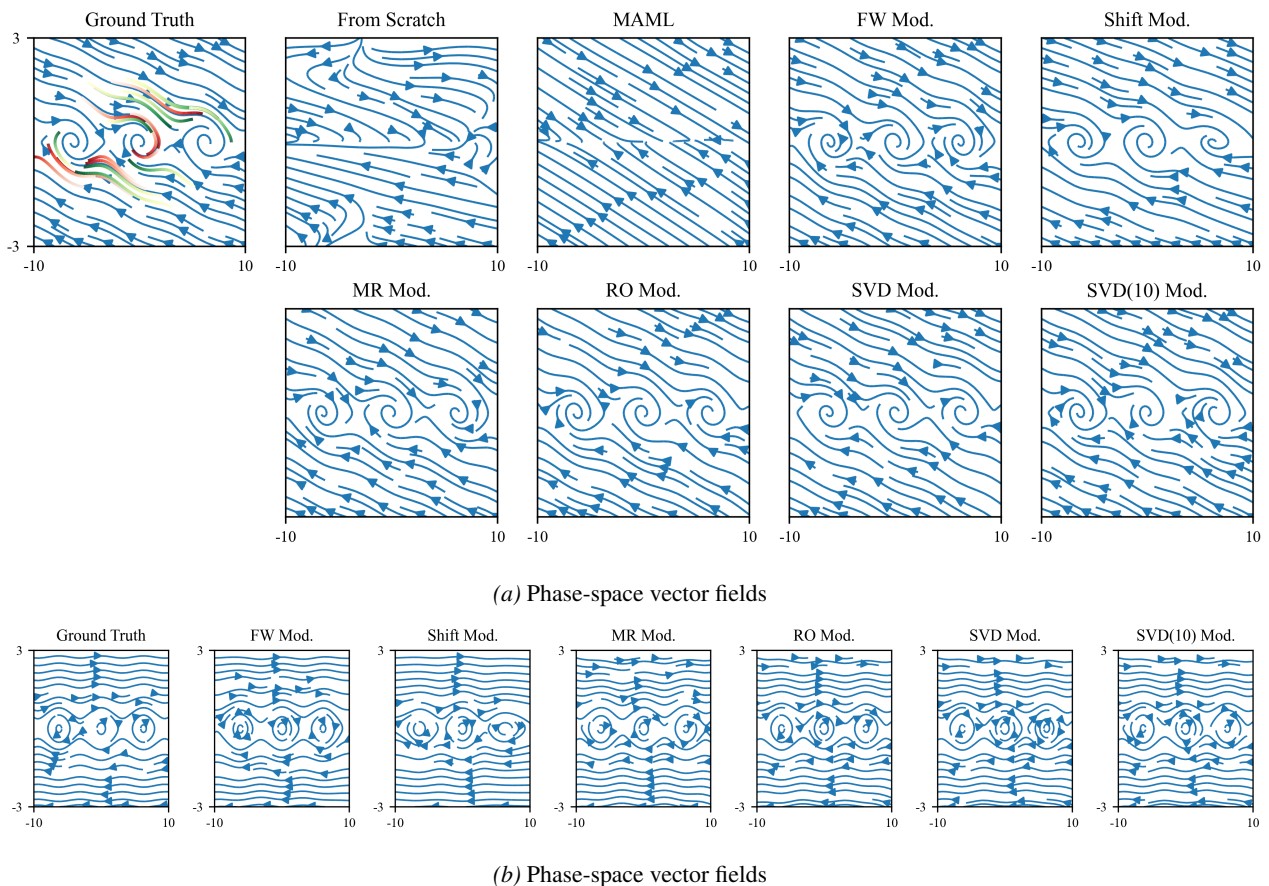

*(a)* Phase-space vector fields

*(b)* Phase-space vector fields

*Figure 24.* [DNO] Illustration of phase space vector fields of the velocity (with $S = 1$, top) and the energy function (bottom) that are obtained through 300-shot adaptation to target fields during the test phase. All fields are depicted on the $(q, p)$ space.

