# OpenReview forum: "Meta-learning Structure-Preserving Dynamics"
_ICML.cc/2026/Conference — ICML 2026 regular_

### Official Review · Reviewer_jZMT · 2026-03-02

**Soundness:** 3
**Presentation:** 3
**Significance:** 3
**Originality:** 3
**Overall Recommendation:** 5
**Confidence:** 2

**Summary:**

The paper addresses the limitation of learning models for discovering physical dynamical systems on a per-configuration basis, which typically requires explicit knowledge of system parameters and costly retraining when these parameters vary. The authors combine structure-preserving neural network architectures — specifically Hamiltonian Neural Networks (HNNs) and GENERIC neural networks (GNNs) — with modulation-based meta-learning.

The proposed framework separates model parameters into shared base parameters and instance-specific parameters generated from low-dimensional latent codes through a hypernetwork. The paper introduces two novel modulation techniques applied layer-wise in an MLP: a multi-rank (MR) modulation inspired by LoRA-style low-rank adaptation, and an SVD-like modulation that decomposes layer updates into learned basis directions with instance-specific coefficients. These are compared against modulation baselines and optimization-based meta-learning approaches (MAML, Reptile, ANIL).

Experiments are conducted on three Hamiltonian systems (mass-spring, pendulum, Duffing oscillator) and one dissipative system (damped nonlinear oscillator), showing that modulation-based approaches — particularly the SVD-like variant — consistently outperform optimization-based baselines in terms of field and trajectory errors.

**Compliance With Llm Reviewing Policy:**

Affirmed.

**Final Justification:**

The reviewers addressed my concerns, and they added a new experiment to reinforce the effectiveness of the method, beyond the simple 2d benchmark originally provided.

**Key Questions For Authors:**

- What do you mean by n-shot adaptation? Do you mean gradient updates, data-points, or trajectories?
- How did you define P in the Hamiltonian of the pendulum? Is the Hamiltonian of the pendulum correct? Should the kinetic energy be $\frac{p^2}{(2ml^2)}$?
- Have you carefully tuned ANIL and Reptile? Do you still have NaN or Div. even after hp tuning?

**Limitations:**

yes

**Strengths And Weaknesses:**

Strengths:
- Novel SVD-like modulation: the paper introduces a novel svd-like modulation which is proven to outperform other methods .
- Systematic study of modulation-based vs optimization-based approaches on three Hamiltonian systems and one dissipative system.

Weaknesses:
- Simple test setting: the experiments are conducted only on 2D dynamical systems.
- Unclear definition of “n-shot” adaptation.

---

> ### Author Rebuttal · Authors · 2026-03-28
>
> We thank the reviewer for the overall positive assessment and provide the following clarifications.
>
> [W1] **simple test setting**
>
> We thank the reviewer for this comment. Due to time constraints, we were able to include one additional experiment beyond the original benchmarks, namely the Kepler system, which represents a more complex dynamical setting compared to the previously considered problems.
>
> We include an additional experiment on the Kepler system, defined in canonical coordinates $x = (q_1, q_2, p_1, p_2)$. The dynamics follow:
>
> $$
> \dot{q}_1 = \frac{p_1}{m_2}, \quad
> \dot{q}_2 = \frac{p_2}{m_2},
> $$
>
>
> $\dot p_1 = -\frac{m_1 m_2 (q_1 - q_{0,1})}{r^3}, \quad
> \dot p_2 = -\frac{m_1 m_2 (q_2 - q_{0,2})}{r^3}.$
>
> where $r = \sqrt{(q_1 - q_{0,1})^2 + (q_2 - q_{0,2})^2}$.
>
> The corresponding Hamiltonian is:
> $$
> H(q,p) = \frac{p_1^2 + p_2^2}{2 m_2} - \frac{m_1 m_2}{r}.
> $$
>
> We vary the system parameters $(m_1, m_2)$ across tasks, sampled from $[0.5, 2.5]^2$.
>
> Here we present experimental results on the above described Kepler problem. Due to the higher dimensionality (4D phase space), we report trajectory error only, as reliable field error estimation would require significantly denser sampling.
>
> **Kepler System (Trajectory Error)**
>
> | Method   | Trajectory Error ↓ |
> |----------|--------------------|
> | Scratch  | 0.4573 |
> | MAML     | 0.2034 |
> | ANIL     | 0.3551 |
> | Reptile  | 0.4558 |
> | FW       | 0.0770 |
> | Shift    | 0.1775 |
> | MR(5)    | 0.0814 |
> | RO       | 0.0635 |
> | SVD      | **0.0516** |
> | SVD(5)   | 0.0548 |
>
> *Results are averaged over a sweep of locality regularization $\{10^{-1},10^{-2},10^{-3}\}$ and weight decay $\{10^{-2},10^{-3},10^{-4}\}$, with 5 runs per configuration.*
>
> [W2/Q1] **"n-shot" definition**
>
> We thank the reviewer for the question. In our setting, “n-shot” corresponds to the number of adaptation steps $N_{\text{test}}$ used during test-time latent optimization in Algorithm 2.
>
> [Q2] **definition of Hamiltonian for pendulum**
>
> We thank the reviewer for pointing this out. The reviewer is correct. This is a typographical error. Our implementation uses the correct expression for the kinetic energy, consistent with the standard Hamiltonian formulation. We will correct the notation in the revision.
>
>
> [Q3] **ANIL/Reptil finetune**
>
> We thank the reviewer for this question. To ensure a fair comparison, we used the same network architecture and training protocol across all methods, without introducing method-specific modifications. As a result, the primary hyperparameter we varied was the learning rate (1e-1, 1e-2, 1e-3), which is the main optimization parameter that can be adjusted consistently across methods.
>
> Under this setting, we did not observe stable or improved performance for ANIL and Reptile, and in some cases training remained unstable (e.g., NaN or divergence). We emphasize that our goal was to evaluate all methods under a consistent and comparable configuration rather than perform method-specific hyperparameter optimization.
>
> We will clarify this in the revision and explicitly state the extent of hyperparameter tuning performed.

---

> > ### Author Rebuttal · Reviewer_jZMT · 2026-04-02
> >
> > The rebuttal fully addressed my concerns, and I increased the score by one point.

---

> > > ### Author Response · Authors · 2026-04-04
> > >
> > > We appreciate the reviewer’s updated assessment and will incorporate the feedback in the revision. Thank you!

---

### Official Review · Reviewer_PviY · 2026-03-08

**Soundness:** 3
**Presentation:** 4
**Significance:** 3
**Originality:** 3
**Overall Recommendation:** 3
**Confidence:** 5

**Summary:**

This paper studies meta-learning for structure-preserving dynamical systems, focusing on Hamiltonian and metriplectic formulations, models where physical structure such as energy conservation or thermodynamic consistency must be preserved. The authors investigate whether modulation based meta-learning can provide an efficient alternative to optimization based meta-learning methods for adapting structure preserving neural networks to families of dynamical systems whose parameters vary across tasks. The proposed approach introduces two modulation mechanisms: multi rank (MR) and SVD like modulation, which adapt a shared base network to different dynamical systems through low rank structured updates generated by a hypernetwork. These approaches are integrated into Hamiltonian and GENERIC neural network frameworks and evaluated on several benchmark dynamical systems like mass-spring, pendulum, Duffing oscillator and a dissipative nonlinear oscillator. Experimental results suggest that modulation based approaches (particularly the SVD modulation variant) outperform optimization based meta learning baselines by improving adaptation accuracy and parameter efficiency while maintaining the structural properties of the learned dynamics.

**Compliance With Llm Reviewing Policy:**

Affirmed.

**Final Justification:**

I thank the authors for the additional clarification. The rebuttal acknowledges that the proposed approach does not provide a formal theoretical justification for the use of low-rank modulation, and instead relies on empirical evidence, heuristic arguments, and capacity considerations. While these observations are useful, they do not establish that low-rank modulation constitutes an appropriate or well-justified inductive bias for families of dynamical systems.

Regarding interpretability, the authors confirm that the latent representation is not explicitly aligned with the underlying physical parameters, and that such alignment is not guaranteed. This reinforces the concern that task-to-task variation is captured through a largely unconstrained latent representation, limiting interpretability in scientific settings.

Overall, while the empirical results are promising, the core assumptions of the method remain insufficiently justified from a theoretical standpoint, and my assessment remains unchanged.

**Key Questions For Authors:**

Question 1: Can the authors provide any theoretical argument supporting why  the proposed method should  work beyond the tested benchmarks?

Questions  2-5: (Justification of low-rank modulation as an inductive bias) The proposed architecture represents task-specific models by combininng shared based parameters with structured perturbations generated from a low-dimensional latent code. Concretely, if $\Theta\subset \mathbb{R}^p$ denotes the parameter space of the neural network representing the Hamiltonian $H_{\Theta}$, the method constructs task-specific parameters of the form $\theta (z)=\theta_{\text{base}}+\phi(z)$; where $z\in \mathbb{R}^d$ is a latent code with $d<<p$ and $\phi(z)$ produces structured low-rank perturbations of the base network weights (i.e. $U(z)V(z)^T$). The set of realizable models induced by the architecture therefore is $\mathcal{S}=\\\{\theta(z): z\in \mathbb{R}^d\\\}\subset \Theta.$ From a mathematical point of view, this means that the architecture restricts the model family to the image of a $d$-dimensional parametrization in the Hilbert space $\Theta$ endowed with the standard Euclidean topology. However, the paper does not provide a justification for why this particular structural restriction should be appropriate for families of dynamical systems. Therefore, it would be useful to clarify the following Questions 2 to 5:

Question 2: The proposed architecture implicitly assumes that the parameters required to represent different tasks can be well approximated by elements of the restricted family $\mathcal{S}$. Is there any structural reason to expect that the set of task-especifc models $\mathcal{S}\subset \Theta$ admits a low dimensional approximation in the Hilbert space topology of $\Theta$ (for example, in terms of small Kolmogorov $n$-width or proximity to a finite-dimensional manifold?).

Question 3: In regard to any relation to the geometry of the induced dynamics: Can this low-dimensional parametrization be related (and precisely how) to properties of the corresponding family of Hamiltonian vector fields in a suitable function space (for example, $L^2$ or Sobolev spaces over phase space)? In other words, is tehre any geometric or functional-analytic property of these dynamical systems that would justify representing task variations through low-rank perturbations of the network weights?

Question 4: Concerning the choice of low-rank perturbations: Why should low-rank perturbations be expected to provide an effective representation of task variations compared to other structured parametrizations? In particular, is there any principled reason to believe (beyond empirical performance) that such perturbations capture an optimal mode of variation across the family of dynamical systems considered?

Question 5: Concerning the optimality of low-rank perturbations: Under what conditions would low-rank perturbations of the base network weights provide an optimal approximation strategy among structured adapters for representing families of dynamical systems? Additionally, is there any characterization of regimes in which low-rank adaptations minimize approximation error or parameter complexity relative to alternative structured parameterizations?

Clarifying Questions 2 to 5 would help determine whether the effectiveness of the proposed modulation mechanism reflects an underlying structural property of the problem or is primarily an empirical observation specific to the benchmark systems considered in the paper.


Question 6: Concerning the lack of interpretability: Since the latent variable $z$ mediates the adaptation across dynamical systems but is not tied to the physical parameters of th system, can the authors characterize the structure of the latent space? For instance, does the learned embedding $z$ admit any relation (topological, geometric or analytic-functional) to the true parameter space of the dynamical systems? Without such a connection, it remains unclear whether the latent representation captures meaningful physical variation or merely acts as an unconstrained optimization variable.


NOTE:  I am not expecting the authors to address the following in the current paper, this is only suggested as a for future work!
 Questions 2 to 5 above concern the structural justification of the proposed model. As a possible direction for future research, it would be interesting to investigate whether any theoretical learning guarantees can be established for the proposed modulation mechanism. For example, one might ask whether MR/SVD modulation can provably recover families of parametrized dynamical systems, generalize across unseen parameter regimes, or guarantee convergence to meaningful latent codes during test-time adaptation? In particular, can one characterize conditions under which the optimization over the latent code $z$ converges (in parameter space) to a model whose induced dynamical system approximates the target system in an appropriate function-space topology (e.g. $L^2$ or Sobolev norms on the vector fields)?

**Limitations:**

"Yes"

**Strengths And Weaknesses:**

Strenghts: The paper addresses an important challenge in scientific machine learning which how to efficiently adapt structure preserving neural networks to families of dynamical systems with varying parameters without retraining a full model for each configuration. The paper introduces two structured weight-modulation strategies (MR and SVD) that provide parameter efficient adaptations of the base model. This proposed modulation based meta learning framework is conceptual simple and integrate naturally with structure preserving models such as Hamiltonian and GENERIC networks while offering a computational efficient approach alternative to optimization-based meta learning methods, potentially enabling faster adaptation in application involving parameter varying dynamical systems. The experimental study includes multiple dynamical systems (both conservative and dissipative), several baselines (MAML, Reptile, ANIL) and multiple evaluation metrics providing consistence evidence of improved performance for the proposed approach.

Weaknesses: My major concerns are the following:

Major Concern 1: The theoretical justification is very limited since the paper does not provide a principled justification for why low-rank modulation should  work beyond the tested benchmarks, so the conclusion relies primarily on empirical evidence. This leads to Question 1 below.

Major Concern 2: The following is one of the paper's weakest conceptual points. The proposed approach relies on low-rank weight modulations as the main mechanism for adapting the model across different dynamical systems. However, it is not clear why low-rank corrections should constitute the appropriate inductive bias for this class of problems, as opposed to other structured parameterizations. It seems to me that the motivation for using low-rank modulation is largely heuristic, in the sense that authors motivation for low-rank updates is mainly by analogy (it seems to me the authors choose low-rank modifications because they are parameter-efficient, easy to optimize and already successful elsewhere)  inspired by techniques such as LoRA and classical SVD decompositions. However, this argument is primarily practical rather than theoretical. Consequently, the choice of low-rank modulation remains an empirically motivated design rather than a theoretically justified one. So, the paper would benefit from clarifying whether there exists a principled reason beyond empirical observations for expecting tsk-to-task variations in the underlying dynamics to admit a low-rank representation. More specifically, can the authors provide a theoretical justification addressing  Questions 2 to 5  below?

Major Concern 3: The proposed method adapts the model tasks through a latent code $z$ that is purely learned through optimization and is not linked to any physical parameters of the underlying dynamical system (so the model does not identify the real system parameters, it only learns a hidden representation that makes the dynamics fit the data). As a result, while the architecture preserves certain structural properties of the dynamics, the mechanism that captures task-to-task variation across systems is effectively a black-box latent representation despite the efficiency advantages of the approach (here efficiency should be intended to the extent of the experimental scope explored by the authors in the paper because of the lack of theoretical justifications mentioned above). So here my main concern is about interpretability, particularly in scientific machine learning settings where people usually seek models that provide insight into the underlying physical mechanisms. This directly motivates Question 6 below.

Minor concerns:

-Minor Concern 1: -the conceptual novelty of the paper is moderate. This is fine but is worth mentioning it explicitly: the proposed approach builds on existing ideas from modulation-based meta-learning and low-rank parameter adaptation, and the main novelty (at least seems to me) lies in applying these ideas to structure-preserving dynamical systems rather than introducing fundamentally new theoretical insights.

--Minor Concern 2: The empirical evidence is promising, however generalization beyond the tests setting is unclear: since the experiments focus on synthetic systems with known governing equations (mostly low-dimensional benchmark families), it is very difficult to assess how robust the approach would be in more realistic scenarios like: higher-dimensional systems, PDEs, chaotic systems, realistic noisy observations, partial observability or model mismatch. Note: while this can be perceived as a Major Concern, I prefer to list it as Minor one expecting the authors to explicitly state in the paper this point as a future work worth deserving attention.



Overall, for the reasons discussed above, I would not recommend acceptance of the paper in its current form. While the empirical results are interesting, the paper lacks a clear theoretical justification for the main claims underlying the proposed approach, namely that task-specific variations in dynamical systems can be effectively captured (optimally) through low-rank perturbations of the network weights (in other words, the paper does not provide a principled explanation for the meta-learning success of the modulation scheme itself). Furthermore, adaptation across systems is mediated through an unconstrained latent code, which limits interpretability and makes it difficult  to understand whether the method captures meaningful system variation or merely fits the training benchmark setup. As a result, without such justifications, it is very difficult to assess whether the observed empirical improvements reflect a genuine structural property of the problem or are specific to the experimental setup considered by the authors.

---

> ### Author Rebuttal · Authors · 2026-03-28
>
> We thank the reviewer for the careful assessment and constructive feedback. We provide the following clarifications.
>
> [W1/Q1] **low-rank assumption**
>
> We appreciate the comment. Please refer to our response to Reviewer Zpf1 comment **[W/Q1]**.
>
> [W2/Q2--Q5] **further theoretical aspects**
>
> We thank the reviewer for these thoughtful questions. We agree that our work does not provide a formal theoretical justification for the specific SVD-like parameterization, nor a proof of optimality among all structured adapters. Our choice is therefore not intended as a uniquely optimal construction, but as a practical instantiation of low-rank adaptation motivated by the low-dimensionality of task variation (as discussed in [W1/Q1]).
>
> The SVD-like parameterization offers several advantages compared to alternative low-rank formulations. First, it allows the effective rank to be determined adaptively: the hypernetwork outputs singular values (with ReLU), which naturally suppress inactive directions and enable instance-dependent rank selection. Second, it provides a decoupled representation between shared basis directions and task-specific coefficients, leading to a more interpretable and stable parameterization compared to unconstrained low-rank factors. Third, it is parameter-efficient at inference time, as only a small number of coefficients (latent variables) need to be optimized per instance, in contrast to multi-rank (MR) formulations that use larger sets of parameters.
>
> Empirically, we observe this strategy is robust to moderate overparameterization (e.g., using a slightly larger maximum rank), with unnecessary directions effectively pruned during training.  This suggests the learned low-dimensional manifold $\mathcal{S}$ is suitably expressive.  This is not entirely surprising, since the dimension of the low-rank correction can easily exceed the dimension of the system parameter space (often small in practice).
>
> We agree that a more formal characterization of when such low-rank parameterizations are optimal, or how they compare theoretically to other structured adapters, remains an open question. We will clarify this point in the revision and position our approach as an empirically validated and practically motivated design choice rather than a theoretically optimal one.
>
> [W3/Q6] **black-box latent code $z$**
>
> We thank the reviewer for this important comment regarding interpretability. In our setting, we explicitly consider the scenario where the underlying physical parameters are unknown and must be inferred from data. The latent variables are therefore introduced as a learned representation to capture this variation.
>
> We agree that the latent variables are not explicitly tied to physical parameters in our formulation, and thus do not provide direct parameter identification. However, we empirically observe that the learned latent space is not arbitrary. As discussed in Appendix G.6, the latent codes exhibit structured variation across systems, and in several cases we observe a consistent geometric relationship between the latent space and the underlying system parameters. This suggests that the latent representation captures meaningful variations in the dynamics rather than acting purely as an unconstrained optimization variable.
>
> That said, we acknowledge that this relationship is not enforced explicitly and may not always be directly interpretable. Incorporating additional structure into the latent space, e.g., via disentanglement or information bottleneck approaches such as $\beta$-VAE style objectives, could potentially improve alignment with physical parameters. We will clarify this limitation and highlight it as an important direction for future work.
>
> Minor comments
>
> [W4] **novelty**
>
> We thank the reviewer for this comment. We agree that our work builds on existing ideas from modulation-based meta-learning and low-rank adaptation. Our primary contribution is to systematically adapt and evaluate these techniques in the setting of structure-preserving dynamical systems, where additional constraints (e.g., conservation laws and dissipation structure) must be maintained.
>
> In this context, the combination of modulation-based adaptation with structure-preserving parameterizations is non-trivial, and our results show that these approaches can be successfully integrated without compromising the underlying physical invariants.
>
> [W5] **generalization to other test setup**
>
> We thank the reviewer for this thoughtful comment. We will explicitly discuss this limitation and highlight it as an important direction for future work in the revised manuscript.
>
> To partially address this concern, we have extended our experiments to include an additional benchmark, namely the Kepler system.
>
> Please refer to our response to Reviewer jZMT comment **[W1]**.
>
> We agree that further evaluation on more realistic settings (e.g., high-dimensional, noisy/partial observation) is important, and we will emphasize this as future work.

---

> > ### Author Rebuttal · Reviewer_PviY · 2026-04-02
> >
> > I thank the authors for the clarifications. The gradient-subspace perspective and additional empirical evidence help motivate the use of low-rank modulation. However, these arguments remain largely empirical and local, and do not provide a principled justification for why low-rank modulation constitutes an appropriate inductive bias. The rebuttal also acknowledges the lack of a formal characterization or optimality result, which reinforces my original concern that the core design choice remains largely heuristic. Similarly, while the latent space appears structured empirically, without an explicit connection its relation to the underlying physical parameters remains unclear, limiting interpretability.
> >
> > Therefore, my overall assessment remains unchanged, 3: Weak Reject.

---

> > > ### Author Response · Authors · 2026-04-04
> > >
> > > We thank the reviewer for the careful reading and responding to our rebuttal.
> > >
> > > We agree that interpretability of the latent representation is desirable, along with establishing an explicit correspondence between the learned latent variables and the underlying physical parameters.
> > >
> > > At the same time, we note that such a correspondence is generally difficult to obtain in latent-variable models trained without explicit supervision. In particular, in autoencoder or auto-decoder (*ours*) frameworks, the latent variables are typically only constrained through reconstruction objectives, and there is no guarantee that they align with the true factors. Moreover, modulation-based meta-learning does not target the parameters themselves, but instead their influence on the network function.
> > >
> > > A substantial body of prior work has studied this problem under the lens of "disentanglement", for example in $\beta$-VAE [1] and its variants [2,3,4], which aim to recover statistically independent latent factors and, thus, improving "interpretability" of the learned latent code. Even in these settings, it is well understood that disentanglement is only achievable under restrictive assumptions, and can break down when the underlying factors are correlated.
> > >
> > > In our setting, the situation is further complicated by the fact that the physical parameters (e.g., $m$ and $l$ in the pendulum) do not necessarily induce statistically independent effects in the observed dynamics (multiplying $m$ by two and simultaneously dividing $l$ by two produces the same dynamics). As a result, we do not expect a simple one-to-one or axis-aligned correspondence between latent variables and physical parameters.
> > >
> > > Nevertheless, we empirically observe that the learned latent space exhibits structured variation across environments, with 96.8% of the variance explained by only five eigenvectors of the Gram matrix $G_{\ell}^T G_{\ell}$.  This suggests that the latent vector captures meaningful system differences rather than acting as a purely unconstrained optimization variable.
> > >
> > > We therefore view explicit alignment between latent variables and physical parameters as an important but separate research direction.  We will be sure that this limitation is clarified in the revision and positioned as future work.
> > >
> > > We acknowledge that we do not have strong theoretical guarantees regarding the use of low-rank modulation. On the other hand, it is reasonable to expect that our low-rank modulations are expressive enough to handle the dynamical systems we consider. Note that a simple counting argument establishes the following: if there are $n$ system parameters $\mu_1,...,\mu_n$ and the rank of the modulation is at least $n$, then the capacity of the class of modulated networks is sufficient to cover variation across the entire parameter space. (Any injective modulation map will work). This intuition is further supported by the effectiveness of low-rank parameterizations in other areas of machine learning, e.g., LoRA in LLMs, low-rankness in (parameterized) physics-informed neural networks, low-rank tensorized kernels in neural operators, etc, which are known to smooth optimization landscapes and mitigate neuron co-adaptation.
> > >
> > >
> > > [1] Higgins et al., *$\beta$-VAE: Learning Basic Visual Concepts with a Constrained Variational Framework*, ICLR, 2017.
> > >
> > > [2] Kim et al., *Disentangling by Factorising*, ICML, 2018.
> > >
> > > [3] Chen et al., *Isolating Sources of Disentanglement in Variational Autoencoders*, NeurIPS, 2018.
> > >
> > > [4] Locatello et al., *Challenging Common Assumptions in the Unsupervised Learning of Disentangled Representations*, ICML, 2019.

---

### Official Review · Reviewer_YvKV · 2026-03-11

**Soundness:** 3
**Presentation:** 3
**Significance:** 3
**Originality:** 3
**Overall Recommendation:** 5
**Confidence:** 3

**Summary:**

To address the limitation that traditional structure-preserving neural networks face high retraining costs when system parameters change, this paper proposes a modulation-based meta-learning framework. By introducing latent-variable multi-rank and SVD-like modulation techniques, the model achieves efficient weight-adaptive updates without requiring explicit prior knowledge of physical parameters.

**Compliance With Llm Reviewing Policy:**

Affirmed.

**Final Justification:**

Applying modulation, particularly SVD-like approaches, to achieve parameter adaptation in physical networks is both insightful and aligned with the interests of SciML.

**Key Questions For Authors:**

First, regarding the fairness of mean-latent initialization: The paper shows that mean-latent initialization outperforms zero initialization, but since the latent space is not subject to an explicit zero-mean prior and instead uses locality regularization, it remains unclear whether zero initialization constitutes a fair baseline. I suggest the authors report the empirical mean of the training latents and clarify whether the advantage of mean initialization persists after adding a centered prior or re-centering the latents.

Second, regarding strict structural guarantees in the dissipative and GENERIC extensions: The extension to dissipative systems is an important contribution of this work. However, after modulating both energy-related terms and the friction core, it is not sufficiently clear whether the degeneracy condition required for thermodynamic consistency is strictly satisfied for arbitrary latents or only approximately satisfied after training. I suggest the authors provide an explicit derivation under this modulated parameterization, or report the degeneracy residual and entropy production after test-time adaptation.

Third, regarding interpretation of the shared-base multi-domain experiment: In the multi-domain experiment, a single shared base network simultaneously fits the pendulum, mass-spring, and Duffing systems. Given that the pendulum uses periodic angular coordinates while the latter two use Euclidean displacement coordinates, it is currently unclear how a shared MLP can uniformly represent these topologically different systems without explicit periodic encoding or manifold modeling. I suggest the authors clarify whether this experiment demonstrates genuine cross-domain structural sharing or primarily successful fitting within limited local regions.

Fourth, regarding pendulum formula and notation consistency: I hope the authors can further clarify the specific definition of the pendulum system. The Hamiltonian form given in the paper does not appear fully consistent with the standard formulation under the usual angular-coordinate and canonical-momentum convention, and the notation in Table 5 also seems inconsistent with the main text. I suggest the authors explicitly define the pendulum state variables and explain whether this stems from reparameterization, notational issues, or other conventions.

Fifth, regarding implementation consistency and fairness of baseline comparisons: Several implementation details require clarification. The main text states that the dynamics MLP uses SiLU, while Appendix F states that the base network uses tanh. Table 1 states that results are averaged over five runs, while Appendix F states that experiments were repeated three times. Additionally, the appendix mentions that Reptile used a different learning rate to avoid divergence. I suggest the authors specify the exact configuration corresponding to the main results and clarify whether all baseline methods received comparable hyperparameter search and stabilization treatment. A concise reproducibility configuration table would be helpful.

Sixth, regarding long-horizon structural preservation after test-time adaptation: The paper emphasizes that the learned dynamics preserve target structure by construction, but the current results focus primarily on field error, trajectory error, SSIM, and few-shot adaptation quality. In contrast, it remains unclear whether structural preservation continues to hold stably during long-horizon numerical rollouts after test-time auto-decoding, especially when using RK4 integration with latents updated from limited samples. I suggest the authors supplement long-horizon evaluations, such as energy drift in Hamiltonian systems and residuals of energy conservation or entropy production over time in dissipative systems, to more directly demonstrate actual structural fidelity after adaptation.

**Limitations:**

Yes

**Strengths And Weaknesses:**

Strengths

The main value of this paper lies in its attempt to answer a practically meaningful question: given a family of dynamical systems that share structural form but differ in specific details, how can one achieve fast adaptation and reasonable structural fidelity without explicitly relying on system parameters? This setting aligns well with common modeling scenarios in scientific machine learning, making the potential applications fairly clear. At the same time, introducing modulation into structure-preserving dynamics modeling provides a natural technical pathway for combining structural constraints with meta-learning adaptation. This direction still carries methodological significance and offers valuable reference for future exploration of more complex, multi-system, and few-shot scientific modeling problems.

Weaknesses

The experimental setup is relatively simple overall, involving mostly low-dimensional, structurally regular classical dynamical models. As a result, the current empirical evidence reads more as a preliminary demonstration of feasibility than as thorough validation of broad applicability. Moreover, the base networks and task distributions employed are not particularly complex, which means the present results do not fully establish whether the method can maintain its advantages on higher-dimensional, more strongly nonlinear, or more heterogeneous systems.

---

> ### Author Rebuttal · Authors · 2026-03-28
>
> We thank the reviewer for the overall positive assessment and provide the following clarifications.
>
> [Q1] **zero init**
>
> We thank the reviewer for this insightful comment. While zero initialization is the standard baseline in prior work, we agree that it may be suboptimal if the learned latent space is not centered. We empirically observe that the mean of the training latent codes is non-zero, and that mean initialization improves performance. Importantly, this improvement persists even when locality regularization is removed, indicating that the benefit is not tied to this design choice but rather reflects better alignment with the learned latent distribution. We will clarify this point in the revision. We will avoid overstating the effect and make explicit that the improvement stems from this distributional shift.
>
> [Q2] **structure-preservation with modulation**
>
> We appreciate the comment. Please refer to our response to Reviewer Zpf1 comment **[Q2]**.
>
> [Q3]  **multi-domain experiments**
>
> We thank the reviewer for this important point. In our implementation, all systems are represented in Euclidean coordinates; in particular, the pendulum is parameterized as $(q, p) \in \mathbb{R}^2$ without explicit periodic encoding. While the pendulum angle is intrinsically periodic, the data are restricted to a single period (e.g., $q \in [-\pi, \pi)$), within which the dynamics are smooth and well-approximated by an MLP.
>
> The goal of this experiment is to evaluate the expressivity of different modulation strategies under heterogeneous dynamics, rather than to claim true cross-domain generalization across differing state-space topologies. We will clarify this point and tone down the wording (e.g., renaming the section title to "multi-domain evaluation") to avoid overstating the scope.
>
> [Q4] **notations**
>
> We thank the reviewer for pointing this out. We clarify that our pendulum formulation follows the standard Hamiltonian neural network (HNN) setup [1], where the state is expressed in canonical coordinates with $q = \theta$ (angular position) and $p = m l^2 \dot{\theta}$ (canonical momentum). Starting from the standard Lagrangian of a simple pendulum, $$ L(\theta, \dot{\theta}) = \frac{1}{2} m l^2 \dot{\theta}^2 - m g l (1 - \cos\theta), $$ the canonical momentum is defined as $$ p = \frac{\partial L}{\partial \dot{\theta}} = m l^2 \dot{\theta}. $$ The Hamiltonian is then obtained via the Legendre transform: $$ H(q,p) = p \dot{\theta} - L. $$ Substituting $\dot{\theta} = \frac{p}{m l^2}$ gives $$ H(q,p) = \frac{p^2}{2 m l^2} + m g l (1 - \cos q), $$ where we identify $q = \theta$. This matches the standard Hamiltonian form used in prior work. Any apparent discrepancy is due only to notational conventions (e.g., setting $m=l=1$ or omitting constants), and we will revise the text and Table 5 to ensure consistency and clarity.
>
> [1] Greydanus, et al, Hamiltonian Neural Networks, NeurIPS, 2019.
>
> [Q5] **inconsistency in setups**
>
> We thank the reviewer for pointing out these inconsistencies. We will correct the manuscript to ensure alignment between the main text and appendix. Specifically,
> * All experiments use SiLU activations for the dynamics MLP
> * The reported results in Table 1 are averaged over five independent runs.
>
> Regarding optimization details, we initialized all methods with a common learning rate. For certain optimization-based baselines (ANIL and Reptile), we observed instability under this shared setting and therefore performed additional hyperparameter tuning (e.g., learning rate adjustment) to ensure stable training. In contrast, the modulation-based methods were stable under the shared configuration and did not require additional tuning. We will clarify these details and include a concise configuration table summarizing hyperparameters for all methods.
>
> [Q6] **long-horizon structural preservation after test-time adaptation**
>
> We thank the reviewer for suggesting this point. While accuracy is uncontrollable out-of-distribution and may be poor, structure-preservation ensures that correct physical behavior is strongly enforced regardless.  This guarantees that solutions exhibit the right conservation and dissipation properties no matter how they are trained, leading to correct thermodynamics.
>
> To numerically verify this, we perform test-time adaptation for DNO, sample a random initial condition, perform long-horizon rollout (with RK4 solver) to produce a trajectory ($\times$ 10 longer than training ones). For all modulation techniques, we observe that $\frac{dE}{dt}$ vanishes to numerical precision (on the order of $10^{-15}$), while $\frac{dS}{dt}$ is nonnegative throughout the trajectories. Please refer to our response to Reviewer Zpf1 comment: **[Q2]** on how to compute these numerically. We will add these in the revision.
>
> ---
> Regarding the concern on the simple test setup, we add one additional test results. Please refer to our response to Reviewer jZMT comment **[W1]**.

---

> > ### Author Rebuttal · Reviewer_YvKV · 2026-04-02
> >
> > I appreciate the authors' explanation and will raise my rating, but I recommend adding a brief discussion in the camera-ready version about whether enforcing a centered prior during training would close the gap between zero and mean initialization.

---

> > > ### Author Response · Authors · 2026-04-04
> > >
> > > We appreciate the reviewer’s updated assessment and constructive feedback, and will incorporate a discussion on enforcing a centered prior during training in the revision. Thank you!

---

### Official Review · Reviewer_Zpf1 · 2026-03-12

**Soundness:** 3
**Presentation:** 3
**Significance:** 3
**Originality:** 2
**Overall Recommendation:** 4
**Confidence:** 2

**Summary:**

This paper studies parameter learning for families of conservative and dissipative dynamical systems. Rather than training a separate model for each parameter configuration, the authors propose a modulation-based meta-learning approach: a shared base neural network with parameters $\Theta_{base}$ is adapted to each instance via instance-specific modulation parameters $\Theta_{indv}$  generated from a latent code.

They introduce two layer-wise modulation schemes. The first is a multi-rank update that adds low-rank corrections to each layer. The second is an SVD-like modulation in which the left/right basis vectors are shared across instances and only the "singular-value" coefficients are produced per instance. Empirically, they evaluate on three conservative systems (pendulum, mass–spring, Duffing) and a dissipative nonlinear oscillator, and find that the SVD-like modulation is often the strongest overall.

**Compliance With Llm Reviewing Policy:**

Affirmed.

**Final Justification:**

I am satisfied by the authors' response; I understand that empirically their method seems to perform well, and that theoretical justification is hard to provide. Therefore, a strong case for acceptance can be made. However, given my limited understanding and background, I am unsure of improving my score.

**Key Questions For Authors:**

1. Can you formalize (even informally) what property of the family $f$ makes shared bases $U,V$ + instance-specific coefficients $d$ sufficient?

2. Since the base architecture is structure-preserving, does modulation ever violate the intended invariants/dissipation properties in practice (numerically)? Any guarantees that the modulated model still respects conservation/dissipation?

3. What regularity do you assume for $f(x, \mu)$ in $x$ and in $\mu$?

4. How do you choose your rank? Do you have guidance on how it should scale with system complexity, dimension?

**Limitations:**

No, please see my weakness subsection.

**Strengths And Weaknesses:**

Strengths:

The proposed layer-wise low-rank modulations (especially the SVD-like variant with shared bases and instance-specific “singular values”) impose a strong inductive bias and reduce per-instance degrees of freedom, which is attractive for data-limited adaptation.

The approach is tested across multiple conservative and dissipative benchmarks, with the SVD-like modulation often performing best overall, including in harder multi-domain generalization settings.

Weaknesses:

While the proposed SVD-like and multi-rank modulations are somewhat intuitive and empirically effective, the paper does not provide a theoretical justification that these parameterizations are expressive enough to capture the variation across system instances. In particular, there is no universal approximation-style result (or even a restricted approximation statement) showing that a family of parameter-dependent vector fields $f(x, \mu)$ can be represented by a shared structure-preserving network plus low-rank, instance-conditioned weight updates, nor any characterization of what classes of parameter variation are well-approximated at a given rank $r$.

As a result, it is difficult for the reader, especially one without strong prior exposure to literature concerning hypernetwork conditioning, to understand why this particular decomposition should be expected to work beyond the reported experiments. The method relies on an implicit low-dimensionality assumption (that instance-to-instance variation differences lie in a small subspace of weight space and can be captured by shared directions with instance-specific singular values), but the paper does not formalize when this holds, how the required rank should scale with system complexity. Adding even a lightweight theoretical perspective would make the motivation substantially less mysterious and improve interpretability.

---

> ### Author Rebuttal · Authors · 2026-03-28
>
> We thank the reviewer for the overall positive assessment and provide the following clarifications.
>
> [W/Q1] **low-rank assumption**
>
> We appreciate the opportunity to clarify the motivation behind our low-rank parameterizations. Beyond practical benefits (parameter efficiency, optimization ease, prior success), there are both theoretical and empirical justifications.
>
> First, we can show under mild assumptions on the velocity field $f(x,\mu)$ that all local variation in the mapping $\mu\mapsto f(x,\mu)$ is of dimension at most $r = \min(n,p)$, where $n$ resp. $p$ are the spatial resp. parametric dimensions. This implies that an $r$-dimensional modulation to the model parameters of $f$ can be sufficient for capturing local parametric variations in $\mu$. This also means that full-rank modulations to the learned $f$ are unlikely to be necessary, which is consistent with the presented work and previous literature. This has been formalized as a Proposition in the revision.
>
> Second, our design is supported by prior theoretical and empirical work [1] showing that task variation in dynamical systems lies in a low-dimensional subspace. Defining the gradient subspace at shared parameters $\Theta_{\text{base}}$ as
>
> $G_{\Theta}$=Span({$\nabla_{\Theta}L(\Theta_{\text{base}}, D^\mu)$}$_{\mu \in P}$),
>
> it is shown that the adaptation directions of linearly parameterized dynamical systems with $n_\mu$ varying physical parameters lie in a low-dimensional subspace, i.e., $ \dim(G_{\Theta}) \le n_\mu \ll \text{dim}(\Theta).$
>
> While derived for linear parameterizations, [1] further shows this persists empirically for nonlinear systems (Appendix D), consistent with observations that gradients concentrate in low-dimensional subspaces.
>
> To further support this, we replicate this analysis by constructing $$
> G = [g_1,\dots,g_{N_{\text{test}}}], \qquad g_i = \mathrm{vec}\left(\nabla_\Theta L(\Theta_{\text{base}}, D^{\mu_i})\right), $$ and examining the spectrum of $G^\top G$.
>
> We observe strong spectral concentration: the top 5 eigenvalues account for approximately 99% of the explained variance. This directly supports that low-rank modulation captures most cross-instance variation. We will include this analysis and clarify that our SVD-like and multi-rank parameterizations are motivated by this gradient-subspace perspective.
>
> [1] Kirchmeyer, et al, Generalizing to New Physical Systems via Context-Informed Dynamics Model, ICML, 2022.
>
> [Q2] **structure-preservation with modulation**
>
> We appreciate the comment. We clarify that our modulation techniques are compatible with Hamiltonian and GENERIC/metriplectic structure-preservation. In these models, the properties necessary for thermodynamics compatibility are not functions of the model parameters: they are hard-constrained for any realization of network weights and biases. Since our modulation operates directly on these parameters, the modulated dynamics also provably retain this underlying structure. We have formalized this as a Proposition in the revision.
>
> For the dissipative (GENERIC) case, the dynamics are given by $$ \dot{x} = L \nabla E + M \nabla S, $$ where $L$ is skew-symmetric and $M$ is positive semi-definite. These structural constraints are enforced at the architectural level and are preserved under modulation, ensuring that the invariants are maintained analytically.
>
> For numerical verification, we compute:
> $$ \frac{dE}{dt} = \nabla E \cdot \dot{x} = \nabla E \cdot L \nabla E + \nabla E \cdot M \nabla S = \nabla E \cdot M \nabla S, $$ $$ \frac{dS}{dt} = \nabla S \cdot \dot{x} = \nabla S \cdot L \nabla E + \nabla S \cdot M \nabla S \geq \nabla S \cdot L \nabla E. $$  The terms on the far right of each line are precisely the obstructions to thermodynamic consistency that provably vanish with the proposed architectures.  We observe that $\frac{dE}{dt}$ vanishes to numerical precision (on the order of $10^{-15}$), while $\frac{dS}{dt} \ge 0$ is nonnegative throughout the trajectories. This numerical validation will be included in the revision and, for better presentation, we will also add visualization of dE/dt and dS/dt.
>
> [Q4] **How to choose rank**
>
> We thank the reviewer for this question. In practice, the rank is chosen as a small upper bound (e.g., 5 or 10) and tuned empirically. We found that using a slightly larger rank does not degrade performance in practice.
>
> In the SVD-like parameterization, the hypernetwork outputs singular values (with ReLU), enabling automatic suppression of unnecessary directions. Thus, the effective rank adapts per instance and is typically lower than the maximum.
>
> Appendix G.5 shows that effective ranks of 3.8, 3.12, and 2.48 depending on the sparsity constraint with similar performance. Additionally, Appendix D shows strong spectral concentration of adaptation directions (e.g., effective rank ≈ 5).
>
> We agree that a more principled scaling of rank with system complexity is an important direction and will clarify this in the revision.

---

> > ### Author Rebuttal · Reviewer_Zpf1 · 2026-04-03
> >
> > While the rebuttal provides helpful intuition (in particular, the gradient-subspace concentration perspective), the first $rank
> > \le \min⁡(n,p)$ point reads as a generic Jacobian-rank observation and does not, on its own, explain why low-rank weight modulation should be sufficient. The gradient-subspace argument is more relevant, but the precise connection between a low-dimensional gradient span and the specific per-layer low-rank parameterization used in the paper would benefit from clearer formalization. Given my limited background in this area, and despite the promising empirical results, I am not comfortable increasing my score beyond my current weak accept based on the current level of theoretical justification.

---

> > > ### Author Response · Authors · 2026-04-04
> > >
> > > We thank the reviewer for the careful reading and for this helpful suggestion. We agree that the connection between the low-dimensional gradient subspace and the specific per-layer low-rank parameterization can benefit from further clarification.
> > >
> > > While a full theoretical characterization is challenging, particularly in the nonlinear setting where such guarantees are generally not available, it is reasonable to expect that our low-rank modulations are expressive enough to handle the dynamical systems we consider. Note that a simple counting argument establishes the following: if there are $n$ system parameters $\mu_1,...,\mu_n$ and the rank of the modulation is at least $n$, then the capacity of the class of modulated networks is sufficient to cover variation across the entire parameter space. (Any injective modulation map will work). Both this and the previous response are not so easily dismissed as a "generic Jacobian-rank observation" when the mapping from latent states to system parameters is implicit in the modulation itself.
> > >
> > > We further aim to provide additional empirical evidence to support this connection.
> > >
> > > Specifically, building on the gradient-subspace perspective, we extend the analysis in a layer-wise manner. That is, instead of considering gradients with respect to all model parameters jointly, we restrict the gradient vectors to the parameter subset of each individual layer and construct the corresponding Gram matrix $G_\ell^T G_\ell$.
> > >
> > > Across all considered systems (mass-spring, pendulum, and Duffing), we observe consistent spectral concentration at the layer level. In particular, the top five eigenvalues account for approximately $96.8$% of the explained variance. This suggests that, even within each layer, environment-induced variation is largely concentrated in a low-dimensional subspace.
> > >
> > > While this observation does not constitute a formal guarantee, it provides empirical support for the use of per-layer low-rank modulation, as it indicates that each layer's parameter updates can be well-approximated within a small number of directions.
> > >
> > > We will include this layer-wise analysis and clarify this connection in the revision.

---

### Decision · Program_Chairs · 2026-04-30

**Decision:**

Accept (regular)

**Comment:**

## Summary
This paper studies parameter learning for families of conservative and dissipative dynamical systems. To address the limitations of traditional structure-preserving approaches such as training on a per-configuration basis, requiring explicit knowledge of system parameters, and costly retraining when these parameters vary, this paper proposes a modulation-based meta-learning framework. By introducing latent-variable multi-rank and SVD-like modulation techniques, the model achieves efficient weight-adaptive updates without requiring explicit prior knowledge of physical parameters.

**The reviewers pointed out the following strengths:**

1. The proposed approach is tested across multiple conservative and dissipative benchmarks, with the SVD-like modulation often performing best overall, including in harder multi-domain generalization settings.

2. Systematic study of modulation-based vs optimization-based approaches on three Hamiltonian systems and one dissipative system.

3. The research direction explored in this paper carries methodological significance and offers valuable reference for future exploration of more complex, multi-system, and few-shot scientific modeling problems.

4. Applying modulation, particularly SVD-like approaches, to achieve parameter adaptation in physical networks is both insightful and aligned with the interests of SciML.
5. **Overall:** The paper solves a problem of significant interest to the AI for Science community, and reviewers have provided positive feedback on its soundness, presentation, and originality.

**The reviewers pointed out the following weaknesses:**

1. The paper lacks a compelling theoretical justification.

2. The motivation for using low-rank modulation is largely heuristic (again no theoretical justification or motivation given in the paper).


## Final justification

The author-reviewer discussion addressed most of the weaknesses in the paper and two of the reviewers raised their score from 4 (weak accept) to 5 (accept). The two reviewers with scores 3 (weak reject) and 4 (weak accept) mainly complain about the lack of theoretical justification. However, the paper has substantial merit and should not be rejected solely on the basis of this weakness. Lastly, as pointed out by the reviewers, adding even a lightweight theoretical perspective would make the motivation substantially less mysterious and improve interpretability.